## OPEN
# Characterization of inositol lipid metabolism in gut-associated Bacteroidetes

Stacey L. Heaver [1], Henry H. Le [2], Peijun Tang[3], Arnaud Baslé[4], Claudia Mirretta Barone[1], Dai Long Vu [5], Jillian L. Waters[1], Jon Marles-Wright[4,6], Elizabeth L. Johnson [2], Dominic J. Campopiano [3] and Ruth E. Ley [1,7] ✉

**Inositol lipids are ubiquitous in eukaryotes and have finely tuned roles in cellular signalling and membrane homoeostasis. In Bacteria, however, inositol lipid production is relatively rare. Recently, the prominent human gut bacterium *Bacteroides thetaiotaomicron* (BT) was reported to produce inositol lipids and sphingolipids, but the pathways remain ambiguous and their prevalence unclear. Here, using genomic and biochemical approaches, we investigated the gene cluster for inositol lipid synthesis in BT using a previously undescribed strain with inducible control of sphingolipid synthesis. We characterized the biosynthetic pathway from myo-inositol-phosphate (MIP) synthesis to phosphoinositol dihydroceramide, determined the crystal structure of the recombinant BT MIP synthase enzyme and identified the phosphatase responsible for the conversion of bacterially-derived phosphatidylinositol phosphate (PIP-DAG) to phosphatidylinositol (PI-DAG). In vitro, loss of inositol lipid production altered BT capsule expression and antimicrobial peptide resistance. In vivo, loss of inositol lipids decreased bacterial fitness in a gnotobiotic mouse model. We identified a second putative, previously undescribed pathway for bacterial PI-DAG synthesis without a PIP-DAG intermediate, common in *Prevotella*. Our results indicate that inositol sphingolipid production is widespread in host-associated Bacteroidetes and has implications for symbiosis.**

nositol, a carbocyclic sugar abundant in eukaryotes, forms the basis for diverse phosphorylated secondary messenger inositol phosphates and inositol lipids. At their simplest, inositol lipids have inositol as their polar headgroup, for example, phosphatidylinositol (PI-DAG; glycerophospholipid backbone; Fig. 1a) or inositol phosphorylceramide (sphingolipid backbone). The inositol headgroup can be further modified, including phosphorylation of PI-DAG to form bioactive phosphoinositides, or addition of a mannose to inositol sphingolipids to form the mannosylinositol phosphorylceramides abundant in yeast[1,2]. Inositol derivatives control key processes of eukaryotic cell physiology. For instance, although phosphoinositides constitute a small fraction of overall phospholipids, they are essential for marking organelle identity, regulating cytoskeleton–membrane interactions, and controlling cell division and autophagy[3,4].

Despite the widespread distribution of inositol lipids in eukaryotes, little is known about the structure and distribution of inositol lipids in Bacteria. Bacterial inositol lipids are comparatively rare and previously thought to be limited largely to PI-DAG synthesis in Actinobacteria (for example, *Mycobacteria*, *Corynebacteria* and *Streptomyces*)[5–7] and Spirochaetes (*Treponema*)[8]. In *Mycobacterium tuberculosis*, an obligate intracellular pathogen, inositol headgroups of PI-DAG link surface oligosaccharide virulence factors to the outer membrane[9].

In eukaryotes, both sphingolipids (SLs, lipids with a sphingosine, or long-chain base backbone) and inositol lipids regulate cell fate and differentiation, inflammation, protein trafficking and gene expression in central metabolic pathways, with imbalances

linked to the pathologies of a growing inventory of diseases[4,10–12]. Inositol SLs, for example, glycosylinositol phosphorylceramides, are abundant in yeast and plants. In Bacteria, inositol SLs are produced by the periodontal pathogen *Tannerella forsythia*[13] (Phylum Bacteroidetes). Recently, Brown et al.[14] reported inositol SLs in the common human gut commensals *Bacteroides thetaiotaomicron* (hereafter BT) and *Bacteroides ovatus*. Ceramide phosphoryl-myo-inositol has also been reported in a free-living Bacteroidetes, *Sphingobacterium spiritivorum*[15], and the Proteobacterium *Myxococcus xanthus*[16]. Whereas inositol lipids and SLs are well studied in plants and fungi, bacterial inositol SL synthesis has been largely overlooked.

Across kingdoms, de novo inositol synthesis begins with the formation of inositol phosphate from glucose 6-phosphate (G6P) by a myo-inositol phosphate synthase (MIPS, Enzyme Commission "EC" # 5.5.1.4)[11]. From here, a bacterial pathway for inositol glycerophospholipid synthesis has been described (for example, in *Mycobacteria*) and differs from the eukaryotic pathway by the direct use of inositol phosphate, not its dephosphorylated inositol form, as a substrate in the formation of PI-DAG. This leads first to the synthesis of phosphatidylinositol phosphate (PIP-DAG) from cytidine-diphosphate-diacylglycerol (CDP-DAG) and MIP, which is subsequently dephosphorylated to PI-DAG[17]. Although the PIP-DAG synthase has been well characterized in Bacteria[18], the phosphatase responsible for the conversion of bacterial PIP-DAG to PI-DAG has yet to be identified[18]. Moreover, although the inositol SL synthesis gene cluster was predicted in BT[14,19], it remains functionally unconfirmed.

[1]Department of Microbiome Science, Max Planck Institute for Biology Tübingen, Tübingen, Germany. [2]Division of Nutritional Sciences, Cornell University, Ithaca, NY, USA. [3]School of Chemistry, University of Edinburgh, Edinburgh, Scotland, UK. [4]Newcastle University Biosciences Institute, Newcastle University, Newcastle-upon-Tyne, UK. [5]Mass Spectrometry Facility, Max Planck Institute for Biology Tübingen, Tübingen, Germany. [6]School of Natural and Environmental Sciences, Newcastle University, Newcastle-upon-Tyne, UK. [7]Cluster of Excellence EXC 2124 Controlling Microbes to Fight Infections, Tübingen, Germany. ✉e-mail: ruth.ley@tuebingen.mpg.de

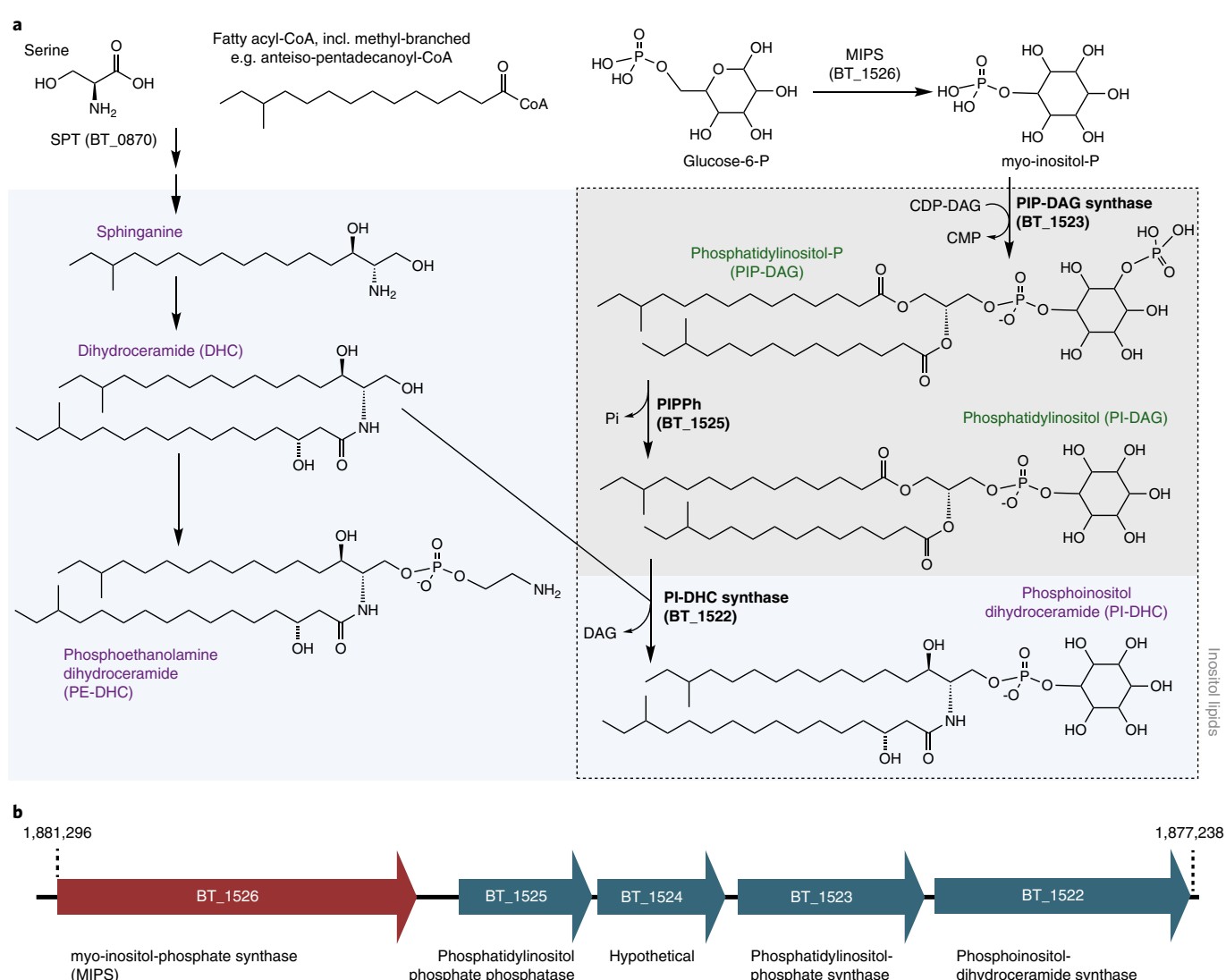

**Fig. 1 | Enzymatic pathway for inositol lipid synthesis in BT. a**, The de novo sphingolipid synthesis metabolic pathway in relation to inositol lipid synthesis, with BT enzymes investigated in this study (bolded black) and representative lipid structures, with chain length and branching reflecting predominant structures determined by fatty acid methyl ester analysis (see Supplementary Information, Supplementary Figs. 1 and 2, and Table 6). Branching patterns of dihydroceramide-based lipids are predicted and not confirmed. Sphingolipid structures are named in purple and shown on a light blue background; glycerophospholipid structures are named in green on a grey background. Inositol lipids are outlined with a dashed line box. SPT, serine palmitoyltransferase. **b**, The genomic region of BT inositol and inositol lipid synthesis, with chromosomal numbering (shown in reverse orientation). Gene colour (blue or red) indicates membership in a predicted operon by BioCyc. Annotations are of the enzyme functions elucidated in this study (due to the lack of a lipid phenotype in its knockout strain, BT_1524 remains hypothetical). Substrates and products for the sphingolipid portion of **a** are not listed, as these reactions (beyond SPT) have not been characterized in the Bacteroidetes.

Here we combine genomic and biochemical approaches to functionally characterize the predicted inositol lipid metabolism gene cluster in BT, from the initial synthesis of myo-inositol-phosphate (MIP) to its addition as a headgroup to glycerophospholipids and SLs. We identify inositol as a component of the cell capsule, and roles for inositol lipids in antimicrobial peptide resistance and bacterial fitness in a mammalian host. Finally, we describe a novel putative alternative gene cluster common in *Prevotella*, revealing an extensive capacity for inositol lipid synthesis among gut-associated Bacteroidetes.

## Results

First, we identified genes responsible for BT inositol lipid metabolism (Fig. 1a). We identified BT_1522 as having high homology to the yeast enzyme inositol phosphorylceramide synthase

(IPC synthase, also known as AUR1) that catalyses the attachment of the phosphorylinositol group onto ceramide (query cover 50%, *e*-value $1 \times 10^{-15}$, percent identity 26%). Thus, we hypothesized that BT_1522 is involved in the synthesis of phosphoinositol dihydroceramide (PI-DHC). *BT_1522* and its gene cluster (Fig. 1b) were previously predicted to be involved in inositol lipid metabolism[14,19]. Adjacent predicted genes in the cluster encode BT_1523 (annotated as a CDP-diacylglycerol-inositol 3-phosphatidyltransferase), BT_1524 (hypothetical protein), BT_1525 (annotated as phosphatidylglycerophosphatase A, PgpA) and BT_1526 (myo-inositol phosphate synthase, MIPS).

We constructed a BT strain with inducible control of the first enzyme in the de novo SL synthesis pathway, serine palmitoyltransferase (SPT; BT_0870; Fig. 1a)[20]. This inducible SPT (iSPT) strain enables

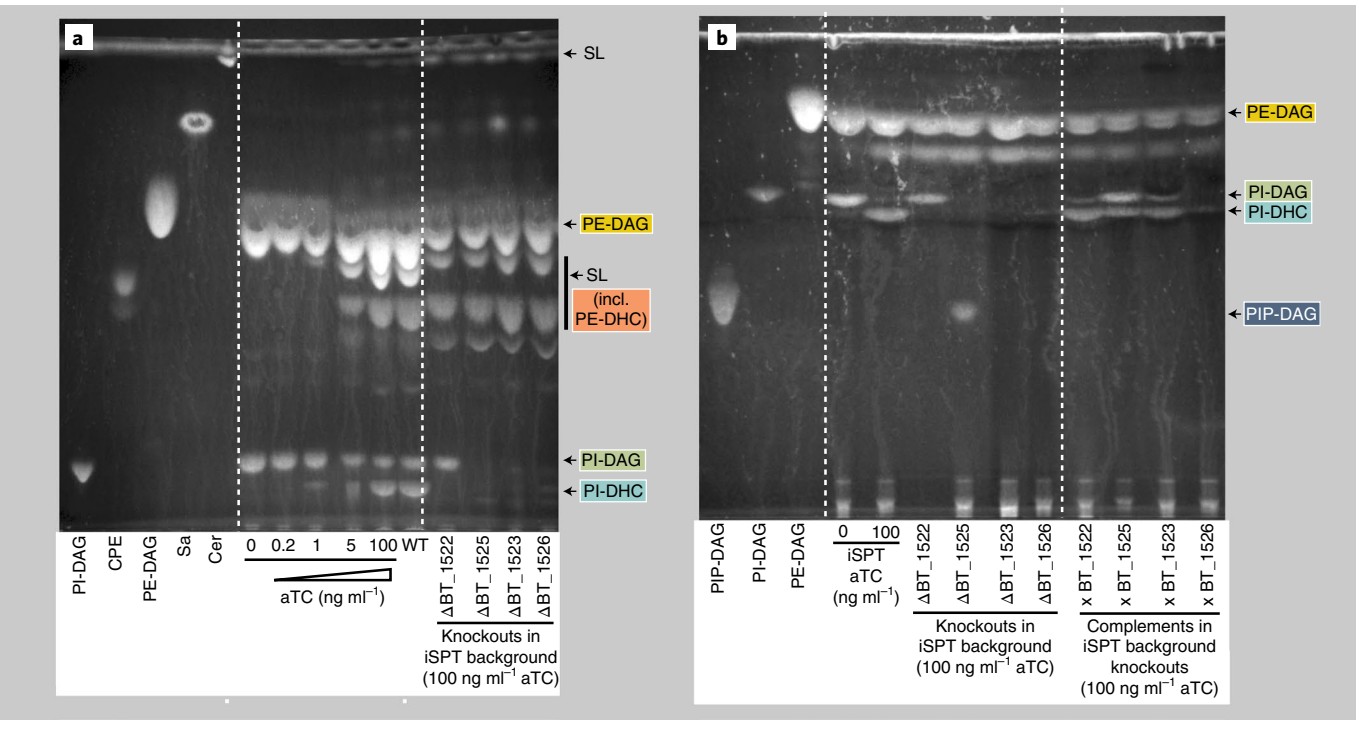

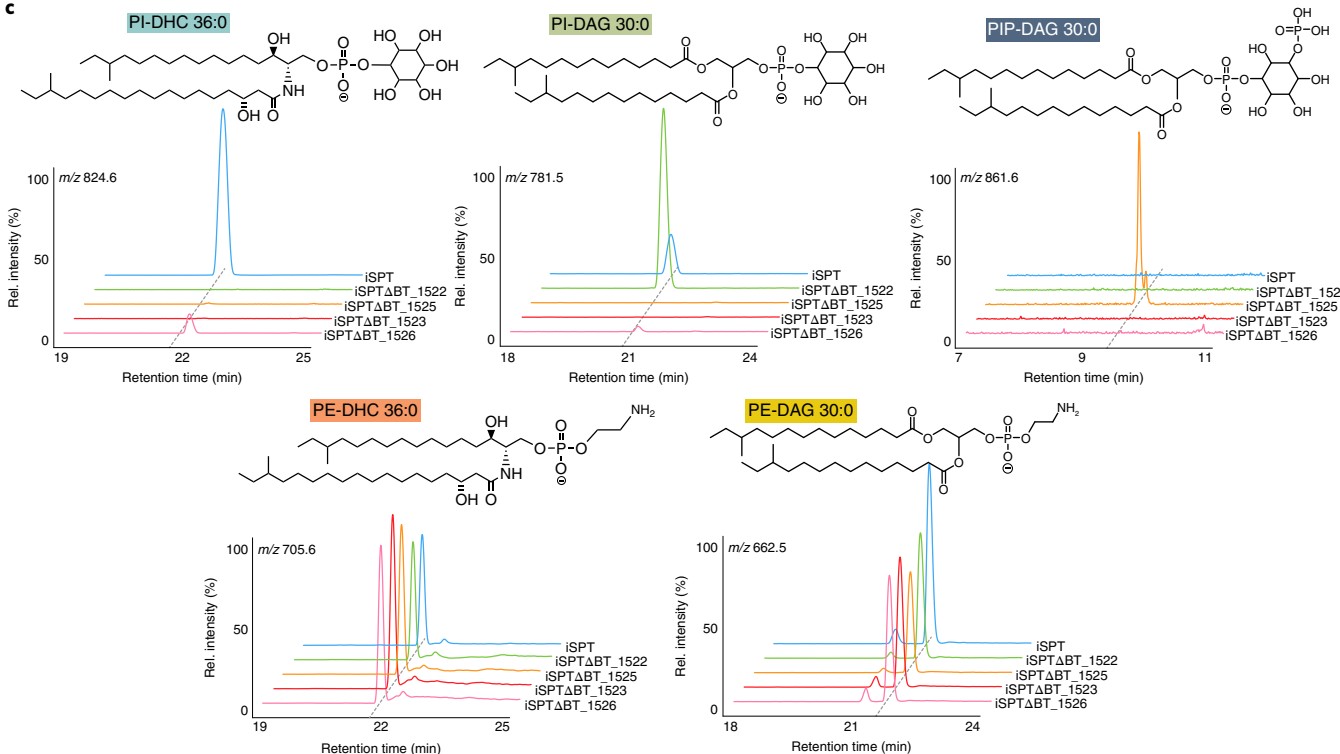

**Fig. 2 | BT produces inositol phospholipids and sphingolipids. a**, TLC of five standards: PI-DAG, 16:0 phosphatidylinositol; CPE, ceramide phosphoethanolamine; PE-DAG, egg yolk phosphatidylethanolamine; Sa, d18:0 sphinganine; Cer, d18:1/18:0 ceramide (left); six standard (non-acidic) lipid extracts from the iSPT BT strain (used as a background for knockout generation) at 0, 0.2, 1, 5 and 100 ng ml⁻¹ aTC induction of SPT, and WT BT VPI-5482 (middle); and standard lipid extraction from ΔBT_1522, ΔBT_1523, ΔBT_1525 and ΔBT_1526 knockout strains in the iSPT background at 100 ng ml⁻¹ aTC induction of SPT (right). **b**, TLC of standards: PI-DAG, PE-DAG as in **a** (left), plus PIP-DAG, 18:1 PI(3)P (left); PIP-DAG lipid extractions of iSPT strains at 0 and 100 ng ml⁻¹ aTC followed by iSPTΔBT_1522, iSPTΔBT_1523, iSPTΔBT_1525 and iSPTΔBT_1526 (middle); and each of their respective complementations at 100 ng ml⁻¹ aTC induction of SPT (right). **c**, Predicted structures and ion chromatograms demonstrating detection of inositol lipids and sphingolipids in iSPT, iSPTΔBT_1522, iSPTΔBT_1523, iSPTΔBT_1525 and iSPTΔBT_1526 at 100 ng ml⁻¹ aTC induction. Branching patterns of DHC-based lipids are predicted and not confirmed.

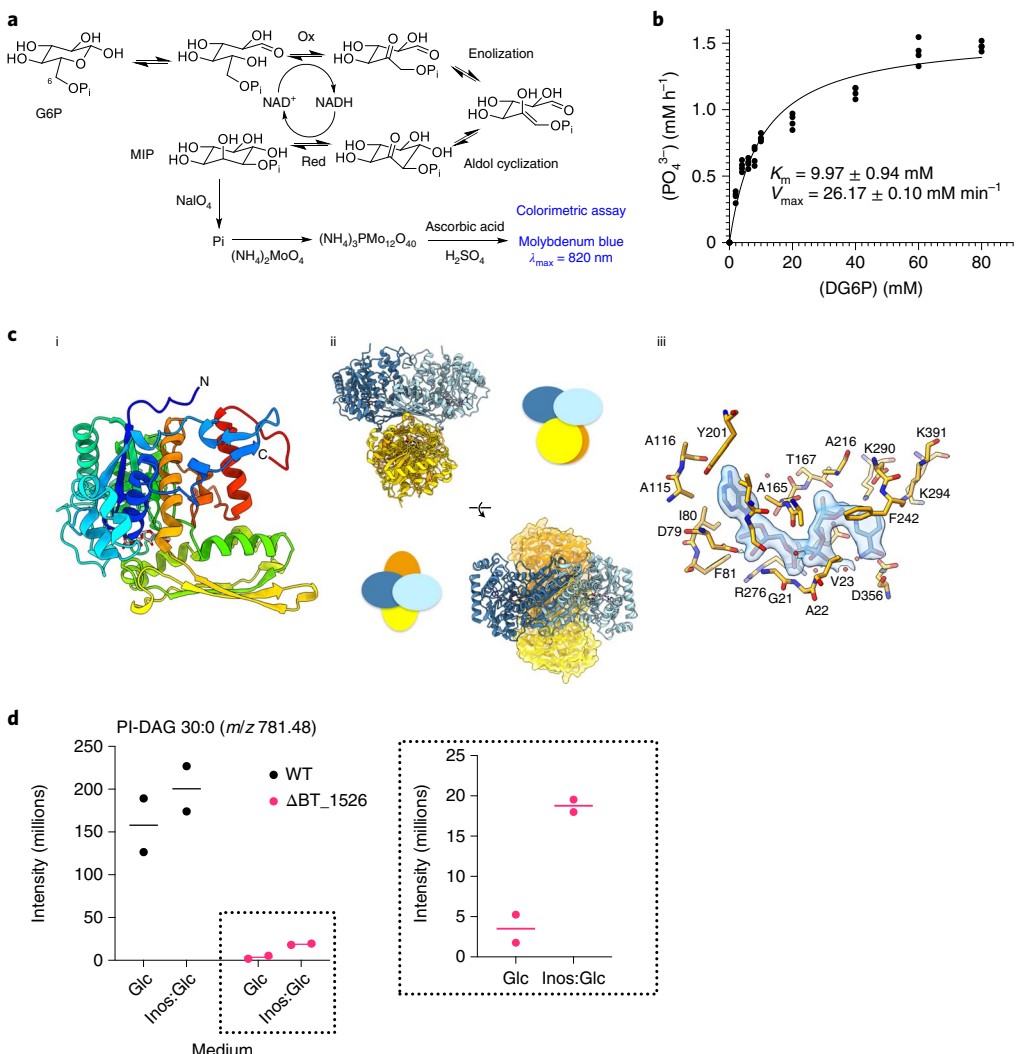

**Fig. 3 | BT_1526 produces myo-inositol-phosphate in vitro. a**, Proposed mechanism for the MIPS-catalysed NAD-dependent/redox-neutral conversion of G6P to MIP. **b**, Molybdenum blue assay for detection of MIP. Kinetic analysis of recombinant BT_1526 MIPS using G6P as substrate. **c**, The crystal structure of BT_1526 MIPS: (i) The monomer subunit, (ii) the tetramer, with cartoon representations to illustrate relative rotations of subunits and (iii) the structure of the MIPS:NAD complex in the cofactor binding site. Letters with numbers indicate the amino acid in the given position in the protein using the one-letter amino acid code. **d**, Production of inositol lipids during growth in inositol-supplemented minimal medium ('Glc', exclusively glucose in medium; 'Inos:Glc', 1:1 molar abundance of inositol:glucose). Intensity values for the lipid peak at $m/z$ 781.48, PI-DAG 30:0, measured in lipid extracts from WT and ΔBT_1526 strains grown either in minimal medium with glucose as the carbon source, or a 1:1 mix of myo-inositol:glucose, with $n = 2$ biological replicates. The inset shows higher $y$-axis resolution for lipid peaks from the ΔBT_1526 strain.

precise control over SL synthesis to produce both PI-DAG and PI-DHC, or solely PI-DAG. As expected, in the absence of SPT induction, we detected no SLs by thin-layer chromatography (TLC) analysis (Fig. 2). Gradual SPT induction led to tunable increases in SL levels, approximating wild-type (WT) SL levels at full induction (Fig. 2a), with SLs comprising ~50% of lipids ($47 \pm 7\%$, $n = 6$). Detected SLs included PI-DHC and phosphoethanolamine dihydroceramide (PE-DHC) (Fig. 2c).

**Functional characterization of the inositol lipid cluster.** To uncover the function of each enzyme in the putative inositol lipid metabolism pathway, we knocked out individual genes (BT_1522 to BT_1526) in the iSPT background by scarless deletion[21] (denoted iSPTΔBT_1522 to iSPTΔBT_1526). Each gene was also knocked out in the WT (Δ*tdk*) background (indicated by for example, ΔBT_1522). We examined the lipid content of the resulting strains (SL synthesis fully induced, unless otherwise stated) using TLC

and high performance liquid chromatography mass spectrometry (HPLC–MS) (see Supplementary Information for more detailed lipid structure analysis). Consistent with the predicted role for BT_1522 as a PI-DHC synthase, the iSPTΔBT_1522 strain failed to produce PI-DHC, but production of PI-DAG and non-inositol SLs, including PE-DHC, was unaltered (Fig. 2a–c, and Extended Data Figs. 1 and 2). Similarly, the iSPTΔBT_1526 strain (lacking the predicted MIPS) failed to produce both PI-DAG and PI-DHC, in accordance with the loss of the myo-inositol-phosphate substrate.

Interestingly, both the iSPTΔBT_1523 and iSPTΔBT_1525 strains also failed to produce PI-DAG and PI-DHC (Fig. 2). The synthesis of other glycerophospholipids was unaffected in the iSPTΔBT_1525 strain, an observation in disagreement with the annotated function of BT_1525 as a PgpA[22]. We hypothesized that BT may use a two-step process to synthesize PI-DAG, similarly to *Mycobacteria*, using a PIP-DAG intermediate[18]. Accordingly, comparison of the functional protein motifs in BT_1523 and BT_1525 with those in the charac-

terized *Renobacterium salmoninarum* PIP-DAG synthase (PIPS)[23] revealed the same conserved catalytic residues (DX₂DGX₂AR…GX₃DX₃D) in BT_1523. This observation supports that BT_1523 functions as a PIPS in the biosynthesis of PIP-DAG.

PIP-DAG has not been reported in BT probably because PIP-DAG extractions under non-acidic conditions are low-yielding. PIP-DAG abundance may also be low in BT, as are phosphoinositides in eukaryotes[3]. Following a PIP-DAG-optimized lipid extraction, we detected high levels of PIP-DAG in iSPTΔBT_1525, which are detectable by both TLC and HPLC–MS (Fig. 2b,c, and Extended Data Figs. 1 and 2). PIP-DAG accumulation in iSPTΔBT_1525 suggests that BT_1525 is most probably a phosphatidylinositol phosphate phosphatase (PIPPh), responsible for the rapid downstream conversion of PIP-DAG to PI-DAG. Although BT_1525 has homology to the phosphatidylglycerophosphatase A protein family (Pfam), this sequence similarity could reflect an expansion of the functional role of this protein motif from the dephosphorylation of a phospholipid glycerophosphate headgroup to an inositol phosphate headgroup. Previous work has shown that transcriptomic expression of *BT_1525* is higher than those of *BT_1523* and *BT_1522* in every growth phase of BT[24], probably enabling the rapid conversion of PIP-DAG to PI-DAG and preventing accumulation of PIP-DAG in BT.

Despite the central location of *BT_1524* in the inositol lipid metabolism gene cluster, inositol lipids in iSPTΔBT_1524 phenocopied BT by TLC (Extended Data Fig. 3). The *BT_1524* gene is predicted to encode an integral membrane protein with a Gtr-A motif (Pfam); other Gtr-A family proteins are involved in cell surface polysaccharide or exopolysaccharide synthesis[25–27].

To confirm that the loss of inositol lipids in knockout strains was not due to off-target effects, we genomically integrated the native BT sequence of each gene into the corresponding knockout strains in the iSPT background (iSPTΔBT_1522, iSPTΔBT_1523, iSPTΔBT_1525, iSPTΔBT_1526), paired with a constitutive promoter optimized for BT[28]. The complementation was successful for three of the four strains (iSPTΔBT_1522, iSPTΔBT_1523, and iSPTΔBT_1525), fully restoring the capacity for both PI-DAG and PI-DHC synthesis (Fig. 2b).

**Structural characterization of BT_1526 MIPS.** To confirm the predicted function of BT_1526 as a redox-neutral, nicotinamide adenine dinucleotide (NAD⁺/NADH)-dependent MIPS, we cloned the gene and heterologously overexpressed the protein in *Escherichia coli* (Fig. 3 and Supplementary Fig. 3). The MIPS activity of BT_1526 was confirmed using a colorimetric endpoint assay (Fig. 3a,b

and Supplementary Information) and the structure resolved to 2.0 Å resolution (Fig. 3c). Outside of the highly conserved ligand binding site, the overall fold and quaternary structure of the representatives of the family in the Protein Data Bank (PDB) is highly conserved, although eukaryotic MIPS proteins have an N-terminal extension that appears to further stabilize the quaternary structure of the protein (Extended Data Fig. 4 and Supplementary Fig. 4). Overall, the enzyme described here as the first MIPS representative in the Bacteroidetes retains the key functional elements of other MIPS enzymes (see Supplementary Information), underscoring its conservation across biological kingdoms.

**Inositol links to the capsule.** Given excess exogenous inositol, BT is unable to take up more than a very minor amount for inositol lipid synthesis (Fig. 3d). To characterize the roles of MIPS and inositol in BT, we analysed the iSPTΔBT_1526 transcriptome at early stationary phase. Twenty-nine genes were differentially expressed between iSPTΔBT_1526 and iSPT. Unexpectedly, the majority were involved in capsule biosynthesis (Supplementary Table 3). Indeed, while SL induction level had little effect on expression of capsular polysaccharide synthesis (CPS) loci in iSPT, iSPTΔBT_1526 had notable upregulation of CPS1 and CPS6 (Fig. 4a). iSPTΔBT_1526 is deficient in inositol lipids (PIP-DAG/PI-DAG/PI-DHC) and any other possible inositol-containing molecules, such as cell surface polysaccharides and components of the capsule. Interestingly, scanning electron microscopy (SEM) revealed heterogeneous capsule structures in iSPT and iSPTΔBT_1526 strains, with more cells of the iSPTΔBT_1526 strain exhibiting a dense exopolysaccharide-like structure connecting adjacent cells (Extended Data Fig. 5).

Transcriptional responses in ΔBT_1522 (lacking PI-DHC) showed no upregulation of CPS1 and CPS6, suggesting that the effect of MIPS deficiency alone caused the capsule effects in the iSPTΔBT_1526 strain. Compared with WT, ΔBT_1522 differentially expressed 37 genes unrelated to the capsule (Supplementary Table 4), indicating that CPS regulation is related to levels of inositol phosphate, but not inositol SLs. These genes encode many hypothetical proteins and membrane-associated proteins, including those involved in carbohydrate metabolism, such as SusD starch-binding protein homologues (BT_3025 and BT_2806). Pathway enrichment analysis revealed an enrichment in ΔBT_1522 of transcripts involved in sugar degradation (specifically, 5-dehydro-4-deoxy-D-glucuronate degradation, $P = 0.006$, Fisher exact test, Benjamini-Hochberg correction), acetate and ATP formation from acetyl-CoA ($P = 0.098$), and carboxylate degradation ($P = 0.098$).

**Fig. 4 | Inositol synthesis influences CPS loci expression and inositol lipids alter AMP resistance and in vivo fitness. a**, Gene expression data (normalized log₂ expression values, scaled by row) in the 8 BT CPS loci. Genes were filtered to include those in which maximum log₂-normalized expression is >1.5 and exclude those with maximum absolute log₂-fold-change difference in expression <1.5 in all pairwise comparisons of conditions. The tree above the expression data represents the Euclidean column clustering defining the sample order by expression similarity. Colour in the far right column indicates gene assignment to one of 8 CPS loci. Strains tested include WT BT, iSPT, ΔBT_1522 and iSPTΔBT_1526 in the iSPT background. SPT induction in the iSPT strains at 0, 0.2, 1.0, 5.0 or 100 ng ml⁻¹ aTC induction is indicated in shades of grey. Labels below each column indicate strain, induction level and replicate ID according to the following pattern: '(strain) - (aTC induction in ng ml⁻¹) (replicate A/B)'. Blue and pink shading emphasizes replicates from the ΔBT_1526 and ΔBT_1522 strains, respectively. **b**, Percent abundance of inositol among total glycosyl residues detected in each WT and inositol lipid knockout strain (n = 3 biological replicates per strain tested; means ± s.d.). Legend and data colours are the same as in **c** and **d**. Full capsule analysis (lipids and glycosyl residues) are available in Extended Data Fig. 6. **c**, IC₅₀ (n = 2 biological replicates per strain) for each WT and BT inositol lipid knockout strain in minimal medium supplemented with the cationic antimicrobial peptide LL-37. Data are representative of two experiments and represented as mean ± s.d. One-way ANOVA $F_{(5,6)} = 35.5$; $P = 0.0002$; Tukey's multiple comparisons: *$P \leq 0.05$, **$P \leq 0.01$, ***$P \leq 0.001$. WT vs ΔBT_1523 $P = 0.0124$; WT vs ΔBT_1525 $P = 0.0007$; WT vs ΔBT_1526 $P = 0.0016$. **d**, Caecal BT colonization of mice after 14 d bacterial association. Copies of BT (c.f.u.-equivalents quantified by qPCR analysis of *BT_1521* copy number) present in mouse caecal contents after 14 d strain association. One-way ANOVA $F_{(3,28)} = 5.6$; $P = 0.004$; Tukey's multiple comparisons: *$P \leq 0.05$, **$P \leq 0.01$ (n = 8 mice; each point is the mean of 3 technical replicate measurements). WT vs ΔBT_1526 $P = 0.0496$; ΔBT_1522 vs ΔBT_1523 $P = 0.0480$; ΔBT_1523 vs ΔBT_1526 $P = 0.0050$. Normality of data distribution was confirmed using the D'Agostino-Pearson test at alpha = 0.05. **e**, Percent abundances of WT in the combined WT + ΔBT_1523 population in the initial inoculum (n = 1, mean of 3 technical replicates) and in the caeca of 8 gnotobiotic mice following a 14 d colonization (n = 8 mice; each point is the mean of 3 technical replicates per mouse). The black line indicates the median of the WT BT abundance in mouse caeca following the competition experiment (73.0%).

Inositol has not been previously reported as a component of BT capsule[29], perhaps due to its common use as an internal standard in the high-performance anion-exchange chromatography with pulsed amperometric detection analysis of capsule components. Using an alternative standard in gas chromatography–mass spectrometry (GC–MS), we detected very low or no inositol in ΔBT_1526's capsule (that is, 0.01% molar abundance in one replicate of ΔBT_1526; Fig. 4b), yet detected minor amounts (at most 0.2% molar abundance) in the capsular glycosyl residues

of WT, ΔBT_1522, ΔBT_1523, ΔBT_1524 and ΔBT_1525 strains. Apart from inositol, the composition of 10 other glycosyl residues in the capsule extracted from each knockout strain (for example, glucose, rhamnose, mannose and so on) phenocopied WT (Extended Data Fig. 6). For the ΔBT_1526 strain, we observed a transcriptomic shift in CPS loci expressed to CPS1 and CPS6 (Fig. 4a), which may be expected to alter the glycosyl residue composition of the capsule[29]; however, this was not observed (Extended Data Fig. 6).

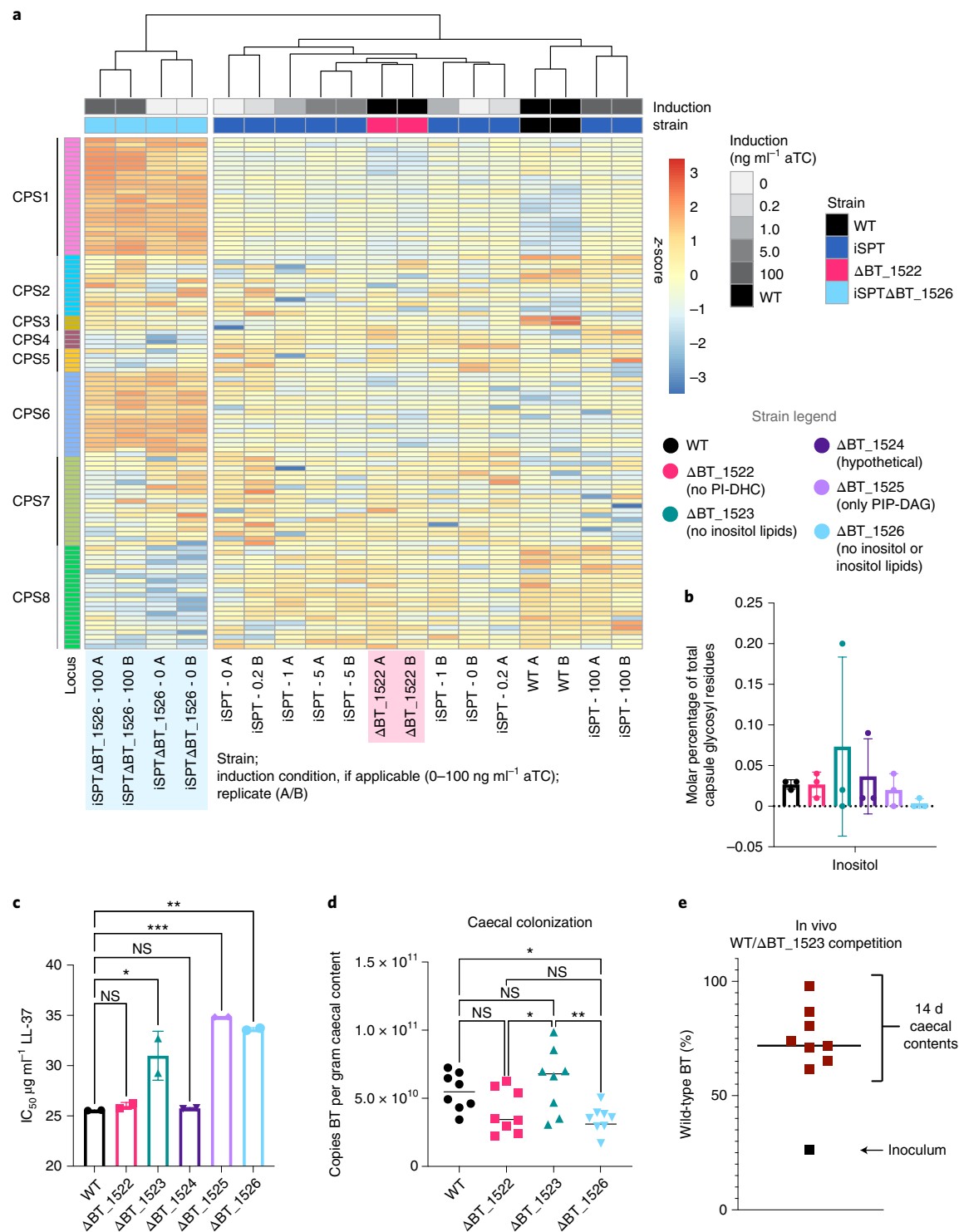

**Inositol lipid deficiency affects bacterial fitness.** To assess the role of inositol and inositol lipids on bacterial physiology, we compared growth of each BT knockout strain. In rich medium, the different knockout strains had similar growth characteristics. In minimal medium with glucose as the sole carbon source, ΔBT_1526, in which MIP cannot be synthesized de novo, had lower cell density at stationary phase (Extended Data Fig. 7).

Alterations to the capsule and cell membrane can be expected to change bacterial susceptibility to antimicrobial peptides (AMPs) that target these structures. To test whether the presence of inositol and inositol lipids altered AMP resistance, we treated each strain with human cathelicidin LL-37, a cationic peptide expressed in the colon that is electrostatically attracted to negative membrane charges (for example, from inositol lipid and phosphatidylserine headgroups)[30]. Consistent with a change in membrane charge, ΔBT_1523 and ΔBT_1526 had a higher half maximal inhibitory concentration ($IC_{50}$) for LL-37 than WT (Fig. 4c). However, ΔBT_1525 was most resistant to LL-37, possibly due to unanticipated effects of PIP-DAG accumulation.

To determine whether inositol and/or inositol lipids are important for fitness in a mammalian host, we mono-associated germ-free (GF) mice with WT, ΔBT_1522, ΔBT_1523 and ΔBT_1526. Each strain was administered to eight 4–6-week-old female C57Bl/6 GF mice fed a standard chow diet ad libitum for 14 d. Previous studies have shown that BT reaches peak colonization levels of $10^{10}–10^{11}$ cells per ml in the caecum after a 10 d colonization[31], therefore we measured caecal cell density following 14 d as a measure of bacterial fitness. We observed a reduction in fitness for ΔBT_1526 compared with WT (Fig. 4d). To test for relative fitness, we competed WT and ΔBT_1523. We chose ΔBT_1523 to specifically test fitness effects of inositol lipids in vivo. GF mice were inoculated with 1:3 WT:ΔBT_1523, but despite its greater proportion in the inoculum, ΔBT_1523 had a clear fitness defect after 14 d, with an average abundance of 76% of the WT strain ($n = 8$, median 73%, range 62–98%; Fig. 4e).

**Inositol lipid synthesis is widespread in gut Bacteroidetes.** To survey for inositol lipid synthesis in the Bacteroidetes, we searched for homologues of MIPS, PIPS, PIPPh, PI-DHC synthase, and the BT InsP6 phosphatase, MINPP[32] (BT_1526, BT_1523, BT_1525, BT_1522 and BT_4744, respectively) in the genomes of 10 species (Fig. 5a and Supplementary Table 5). TLC analysis of lipids from these species revealed that most had lipid bands consistent with PI-DAG and/or PI-DHC in agreement with their genomically-predicted capacity (the exception was *Flectobacillus major*). However, we were surprised to also observe lipid bands consistent with the synthesis of PI-DAG and PI-DHC in species lacking homologues of BT_1522/23/25. HPLC–MS analysis of these lipids confirmed that two species genomically predicted to lack inositol lipids (*B. vulgatus* and *P. veroralis*) produced PI-DHC (Fig. 5b).

To understand why bacterial species lacking BT inositol lipid synthesis homologues nevertheless produced these lipids, we searched the genomes of related species containing a BT_1526 (MIPS) homologue but lacking the remainder of the BT cluster. Using PHI-BLAST with the conserved catalytic residues in BT_1523 (DX₂DGX₂AR…

$GX_3DX_3D$)[23], we identified a predicted CDP-alcohol phosphatidyltransferase genomically encoded near the MIPS homologue in *B. vulgatus*. Almost every *Bacteroides/Prevotella* species containing a MIPS homologue has one of two clusters directly in the vicinity of *MIPS*—either the BT-like cluster (*BT_1522/23/25*), or an alternate cluster encoding an NTP transferase (nucleotidyltransferase) domain-containing protein, CDP-alcohol-phosphatidyltransferase, and haloalkanoate dehalogenase (HAD) hydrolase (Fig. 6). The NTP transferase domain family protein (NCBI Conserved Domain Family cl11394) also shares homology with a phosphocholine cytidyltransferase motif, suggesting that this protein may synthesize cytidine 5'-diphosphoinositol (CDP-inositol), similar to the synthesis of CDP-inositol as a precursor to di-myo-inositol phosphate solutes and dialkylether glycerophosphoinositol lipids in hyperthermophiles[33,34]. The HAD hydrolase superfamily is large and diverse, with the majority of characterized members functioning as phosphotransferases[35]. As a lipid phosphate phosphohydrolase, this HAD hydrolase may function similarly to AUR1[36], acting as a PI-DHC synthase such as BT_1522.

The functions of the NTP transferase domain protein, CDP-alcohol-phosphatidyltransferase, and HAD hydrolase are not confirmed but offer an alternative pathway to enable synthesis of PI-DHC without a PIP-DAG intermediate (similar to PI-DAG synthesis in eukaryotes[37]), with PI-DAG synthesis resembling the synthesis of phosphatidylethanolamine or phosphatidylcholine in the Kennedy pathway[38,39]. Following this logic, the NTP transferase protein would first synthesize CDP-inositol from myo-inositol phosphate and CTP. CDP-inositol and a diacylglycerol (DAG) substrate would then be converted to PI-DAG by the CDP-alcohol-phosphatidyltransferase, and PI-DAG would be converted to PI-DHC by the HAD hydrolase (see pathway comparison in Extended Data Fig. 8). The MIPS homologue is most commonly clustered directly with these other genes, with some exceptions (for example, *P. copri*) (Fig. 6). Interestingly, in *P. veroralis*, the CDP-alcohol phosphatidyltransferase and HAD hydrolase proteins are fused (Supplementary Fig. 5), suggesting the possibility for a cohesive single-enzyme conversion of CDP-inositol to PI-DHC through a PI-DAG intermediate. Some additional putative enzymes are shared in the vicinity of both clusters, including those annotated as a lysylphosphatidylglycerol synthase (BT_1521 homologue) and a carboxypeptidase-regulatory-like domain protein (BT_1527 homologue) (Fig. 6). This alternative pathway could explain PI-DHC synthesis by *P. veroralis* and *B. vulgatus* despite their lack of homologues to the BT inositol lipid cluster (BT_1522/23/25).

Among the Proteobacteria species tested, *Sphingomonas paucimobilis* and *Novosphingobium acidiphilum* made only non-hydroxylated SLs (Extended Data Fig. 9). Despite lacking homology to either the BT-like or putative alternative inositol lipid cluster, *N. acidiphilum* produced an SL with a retention time and headgroup fragmentation consistent with *Bacteroides* PI-DHC fragmentation (Extended Data Fig. 9a). *S. paucimobilis* also produced a lipid with fragmentation similar to *Bacteroides* PI-DHC, but lacking the fragment at 241 *m/z*, tentatively suggesting a phosphorylated-hexose DHC unlike those produced by *Bacteroides* (Extended Data Fig. 9b). Definite

**Fig. 5 | The capacity to produce PI-DHC is widespread among sphingolipid-producing bacteria. a**, TLC of lipid standards and lipid extractions from a diverse array of sphingolipid-producing bacteria. Lanes 1–5, left to right (purple background): PI-DAG, CPE, PE, Sa, Cer. From the sixth lane onwards (blue and green backgrounds) are standard Folch (non-acidic) lipid extractions from: BT iSPT 0 ng ml⁻¹ aTC induction (no SL), WT BT, *B. uniformis* (DSM 6597), *B. fragilis* (DSM 2151), *B. vulgatus* H5_1 (DSM 108228), *B. vulgatus* (DSM 1447), *Prevotella veroralis* (ATCC 33779), *Prevotella copri* (DSM 18205), *Porphyromonas gingivalis* (DSM 20709), *F. major* (DSM 103), *A. malorum* (DSM 14337), *S. paucimobilis* (ATCC 29837) and *N. acidiphiulum* (DSM 19966). Homology to BT protein sequences in the inositol lipid cluster using NCBI BlastP (at *e*-values below 0.001) are indicated below species names with a white circle. Bacteroidetes spp. are on a blue background; α-Proteobacteria spp. are on a green background. **b**, Predicted structures and ion chromatograms of PI-DHC structures in lipids extracted from the diverse sphingolipid-producing species. PI-DHC structures include PI-DHC 34:0(OH), PI-DHC 35:0(OH), PI-DHC 36:0(OH), PI-DHC 35:0 and PI-DHC 36:0. Acyl chain length and branching patterns of lipid structures shown are hypothetical, with this distribution probably variable between species.

characterization of this lipid's phosphohexose identity would, however, require purification and analysis beyond the scope of this study. In addition, the TLC analysis (Fig. 5a) shows a lipid band in the PI-DAG/PI-DHC region for mouth-associated *Porphyromonas gingivalis*, which is probably a phosphorylglycerol-DHC (Extended Data Fig. 9c)[40].

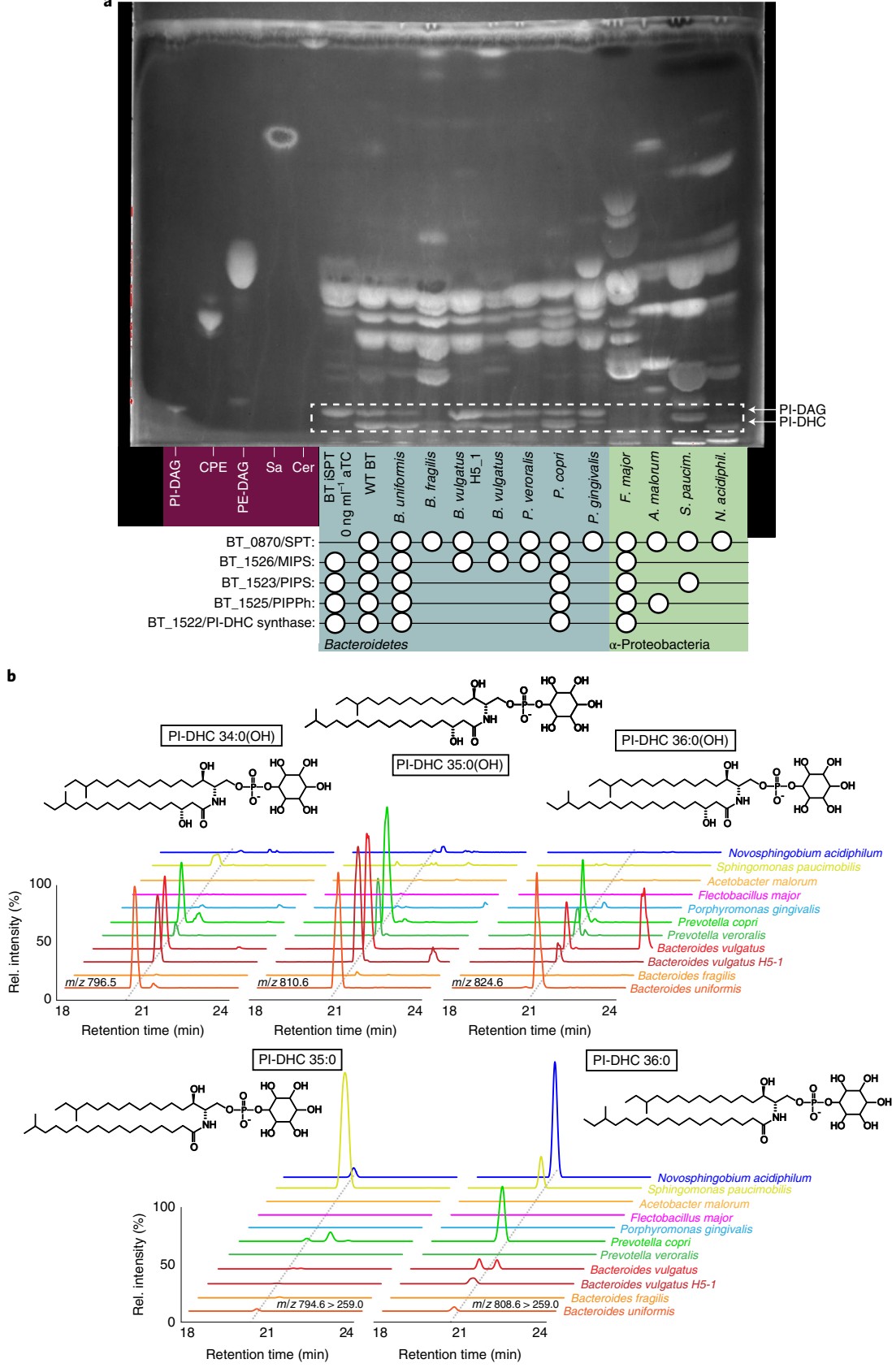

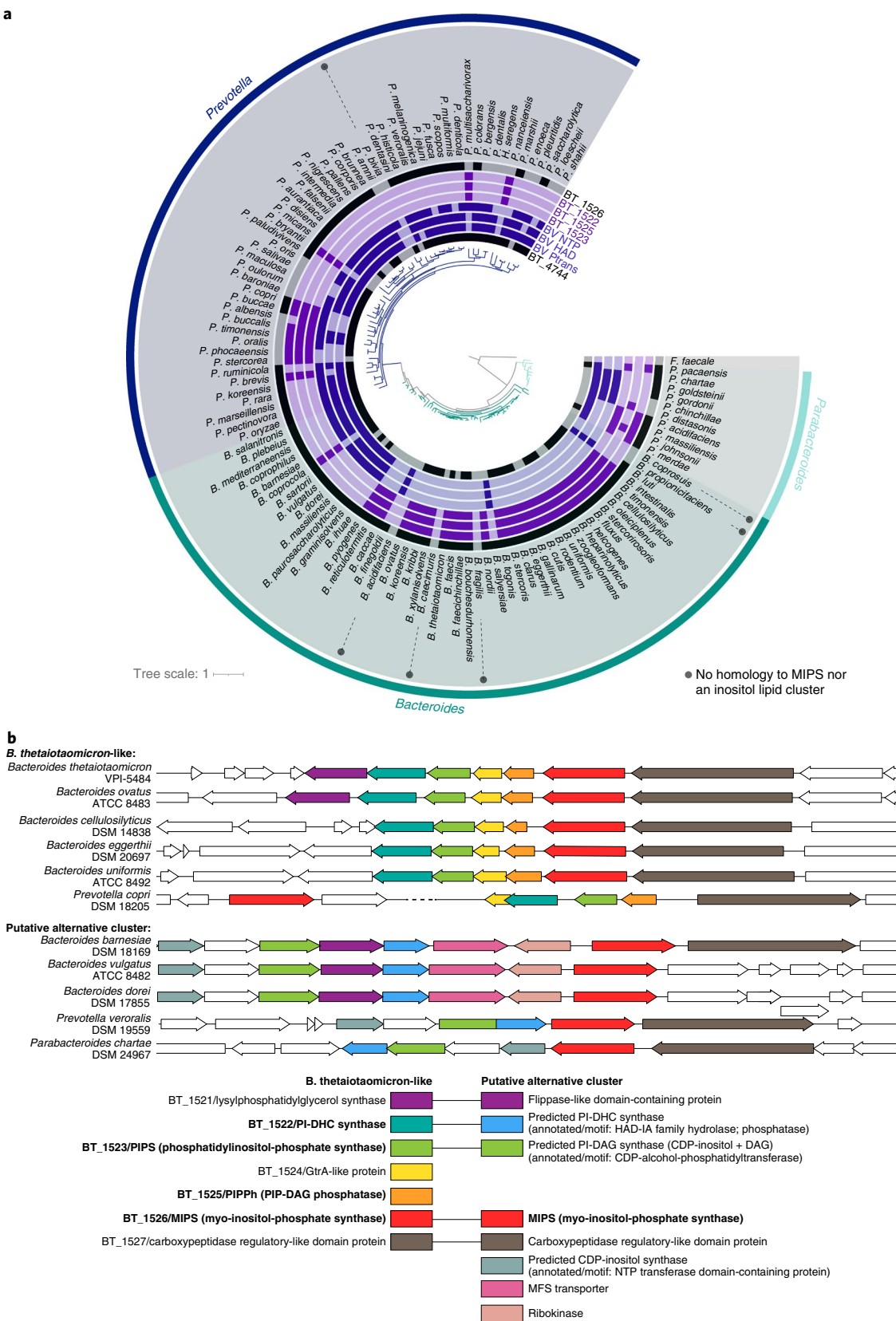

**a**

**b**

B. thetaiotaomicron-like:

Bacteroides thetaiotaomicron
VPI-5484

Bacteroides ovatus
ATCC 8483

Bacteroides cellulosilyticus
DSM 14838

Bacteroides eggerthii
DSM 20697

Bacteroides uniformis
ATCC 8492

Prevotella copri
DSM 18205

Putative alternative cluster:

Bacteroides barnesiae
DSM 18169

Bacteroides vulgatus
ATCC 8482

Bacteroides dorei
DSM 17855

Prevotella veroralis
DSM 19559

Parabacteroides chartae
DSM 24967

B. thetaiotaomicron-like | Putative alternative cluster

BT_1521/lysylphosphatidylglycerol synthase — Flippase-like domain-containing protein

**BT_1522/PI-DHC synthase** — Predicted PI-DHC synthase (annotated/motif: HAD-IA family hydrolase; phosphatase)

**BT_1523/PIPS (phosphatidylinositol-phosphate synthase)** — Predicted PI-DAG synthase (CDP-inositol + DAG) (annotated/motif: CDP-alcohol-phosphatidyltransferase)

BT_1524/GtrA-like protein

**BT_1525/PIPPh (PIP-DAG phosphatase)**

**BT_1526/MIPS (myo-inositol-phosphate synthase)** — **MIPS (myo-inositol-phosphate synthase)**

BT_1527/carboxypeptidase regulatory-like domain protein — Carboxypeptidase regulatory-like domain protein

Predicted CDP-inositol synthase (annotated/motif: NTP transferase domain-containing protein)

MFS transporter

Ribokinase

To assess the distribution of the BT inositol lipid cluster (BT_1522/23/25) and the potential alternative pathway among the Bacteroidetes, we searched for homology in 162 representative species of Bacteroidetes (Supplementary Table 5). Most strains with a homologue of BT_1522, BT_1523 or BT_1525 have homologues of all three, but the distribution does not track phylogeny, supporting lateral exchange among host-associated species (Fig. 6). Roughly three-quarters of *Bacteroides*, *Prevotella* and *Parabacteroides* species have a MIPS homologue paired with either the BT-like inositol lipid cluster or the putative alternative cluster. One notable exception is

**Fig. 6 | The capacity for inositol lipid synthesis is widespread within the Bacteroidetes. a**, Maximum-likelihood-based phylogeny of representative *Bacteroides, Prevotella* and *Parabacteroides* species produced from 71 conserved single-copy genes present in all genomes (identified and concatenated using Anvi'o) and generated by RAxML (best tree; substitution model PROTCAT, matrix name DAYHOFF, Hill-climbing algorithm, bootstrap 50); *Flavobacterium faecale* is included as an outgroup. The rings surrounding the tree indicate species with genes that have NCBI BlastP homology to the BT inositol lipid cluster (in light purple; BT_1522, BT_1523, BT_1525, BT_1526), the BT Minpp (BT_4744), or representative proteins from the *Bacteroides vulgatus* putative alternative inositol lipid cluster (in dark purple; phosphatidyltransferase: BVU_RS13105 'BV Ptrans'; HAD hydrolase: BVU_RS13115 'BV HAD'; NTP transferase: BVU_RS13095 'BV NTP'). Homology at an *e*-value below $1 \times 10^{-8}$ is indicated by dark coloration in the inner circles. **b**, Genomic regions surrounding the BT_1526/MIPS homologue in representative Bacteroidetes, compiled using the PATRIC 3.6.9 Compare Region Viewer. Protein homology (determined using NCBI BlastP) to proteins in the BT-like inositol lipid metabolism cluster (left in key) or the *Bacteroides vulgatus*-like putative alternative inositol metabolism cluster (right in key) is indicated by colour. The functions of enzymes in bold were characterized in this study; sequences with predicted redundant functions between both clusters are linked in the key.

*Bacteroides fragilis*, which does not produce inositol lipids but synthesizes the bioactive glycosphingolipid α-galactosylceramide[41,42]. The BT-like gene cluster is roughly two times more common than the alternative cluster among *Bacteroides* species, while the alternative cluster is approximately four times more common in *Prevotella*. Homologues to proteins in either cluster were absent or highly infrequent in the genera *Porphyromonas*, *Sphingobacterium* and *Chlorobium*, with the exceptions of moderate alternative cluster homology in *Sphingobacteria* and extensive BT_1525 homology in *Chlorobium* (Supplementary Table 5), which may reflect a true phosphatidylglycerophosphatase function.

## Discussion

Inositol lipids have only recently been reported in a few commensal gut bacteria. In this study, we characterized the gene cluster recently hypothesized to be involved in bacterial inositol lipid synthesis in BT to show a functional role for these genes in the Bacteroidetes. BT synthesizes PI-DAG using a mycobacterial-like pathway with a PIP-DAG intermediate; previously, the bacterial PI-DAG synthesis pathway lacked a PIPPh, which we have identified here as BT_1525. We also identified a putative alternative pathway for PI-DHC synthesis, common among *Prevotella* species, that lacks a PIP-DAG intermediate, resembling the eukaryotic Kennedy pathway for phosphatidylethanolamine and phosphatidylcholine synthesis. The majority of host-associated Bacteroidetes species encode one or the other of these pathways, indicating that inositol lipid production is a fundamental trait in the phylum. Together with the importance of inositol lipids in pathogen–host interactions[43] and their impact in this work on fitness in a mammalian host, their high prevalence in the host-associated Bacteroidetes suggests an unexplored role for inositol lipids in commensal–host interactions.

We noted a link between inositol production and the BT capsule, which contained very small amounts of inositol. Importantly, we also detected inositol in capsule from ΔBT_1523, which lacks inositol lipids, indicating that the presence of inositol is not solely lipid-linked. Despite its low capsular abundance, inositol may play an important structural role, as observed in *Mycobacteria*, where >100 glycosyl residues can be bound to a single lipid-linked inositol[44,45]. However, the low inositol abundance in our capsule analysis lacks enough statistical power to determine this unequivocally in BT. The CPS loci expressed by BT, and differentially expressed when MIPS is deficient, have been shown to influence recognition by the host adaptive immune system and bacteriophage[29,46]. The strong effect of MIPS deficiency on transcription of capsule-related genes may therefore indicate an indirect role for inositol on cross-kingdom interactions.

Plants and yeasts also produce inositol-linked SLs critical for fundamental aspects of the organism's physiology, such as protein anchoring and programmed cell death[47]. The yeast homologue of BT_1522 is an antifungal target inhibited by the cyclic depsipeptide natural product aureobasidin (hence the name AUR1)[48]. Although ΔBT_1522 had few overall transcriptomic changes relative to WT controls, the affected pathways are central to carbohydrate degradation

and energy synthesis. This result indicates that similar to these more distantly related organisms, inositol SLs appear to influence pathways central to bacterial physiology.

Our results show that deficiencies in inositol and/or inositol lipids affect interactions with AMPs secreted by the mammalian host and its fitness in vivo. Importantly, deficiency in the ability to produce inositol cannot be rescued by assimilation of inositol from the growth medium and impacts growth both in vitro and in vivo. Taken together, our results indicate that inositol and inositol lipids are probably implicated in resistance to host immune defences through their roles in the structure of the membrane and the capsule, and important for fitness in the mammalian gut.

Our comparative genomic analyses revealed inositol lipid synthesis to be far more widespread in host-associated Bacteroidetes than previously thought. Although the putative alternative pathway remains to be functionally confirmed, the vast majority of species investigated encoded one of the two pathways, with the alternative pathway being more common in *Prevotella*. The extensive prevalence of this function is in agreement with the widespread capacity in gut commensals for phytate (InsP6) degradation, which releases phosphorylated inositol derivatives. Although InsP6 phosphatase is rare across Bacteria (present in only 2.2% of completed genomes in the European Bioinformatics Institute database in 2014), the majority of these enzymes are found in gut microbiome-affiliated species[32]. In addition to the widespread capacity for de novo synthesis of inositol and its lipids reported here, these observations suggest that inositol and inositol lipid cycling in the gut are fundamental attributes of the gut microbiome.

Of the six dominant *Bacteroides* species in the human gut[49], five have genes with homology to the BT-like inositol lipid cluster (*B. cellulosyliticus*, *B. eggerthii* and *B. ovatus*) or the putative alternative cluster (*B. dorei* and *B. vulgatus*), indicating potential for inositol lipid synthesis. As one of the most abundant phyla within the human gut, the widespread synthesis of inositol lipids from gut-associated Bacteroidetes (*Bacteroides*, *Prevotella* and *Parabacteroides* spp.) could represent a significant contribution to the lipid milieu of the gut. Bacterial lipids with high structural similarity to eukaryotic bioactive lipids (for example, SLs) have been shown to influence the metabolism and immune homoeostasis of their hosts[14,20,41,50–52]. Likewise, bacteria are already known to manipulate their host through inositol and inositol lipid metabolic pathways[32,53], and many bacterial and viral pathogens have also adapted to hijack the host phosphoinositide system[54,55]. Our work suggests that inositol and inositol lipid metabolism are prevalent in host-associated Bacteroidetes and represent an unappreciated means of cross-kingdom communication, with effects on the host that remain to be ascertained.

## Methods

**Bacterial strains and culturing conditions.** Unless otherwise stated, all liquid *B. thetaiotaomicron* VPI-5482 Δ*tdk* ('WT BT') cultures were grown anaerobically (95% $N_2$ and 5% $CO_2$ atmosphere) at 37 °C in supplemented BHI media (BHIS;

$37\,g\,l^{-1}$ brain-heart infusion, $5\,g\,l^{-1}$ yeast extract, $1\,mg\,l^{-1}$ menadione, $1\,mg\,l^{-1}$ resazurin, $10\,mg\,l^{-1}$ hemin and $0.5\,g\,l^{-1}$ cysteine-HCl. *E. coli* cultures were grown aerobically at 37 °C in Luria broth with shaking. Final concentrations of antibiotics and selection agents were as follows: erythromycin $25\,\mu g\,ml^{-1}$, gentamicin $200\,\mu g\,ml^{-1}$, streptomycin $100\,\mu g\,ml^{-1}$, carbenicillin $100\,\mu g\,ml^{-1}$ and 5-fluoro-2'-deoxyuridine $200\,\mu g\,ml^{-1}$. In select experiments, BT was grown in *Bacteroides* minimal media (BMM); per litre: $13.6\,g\,KH_2PO_4$, $0.875\,g\,NaCl$, $1.125\,g\,(NH_4)_2SO_4$, 5 g glucose (pH to 7.2 with concentrated NaOH), 1 ml hemin (500 mg dissolved in 10 ml of 1 M NaOH then diluted to final volume of 500 ml with water), 1 ml $MgCl_2$ (0.1 M in water), 1 ml $FeSO_4{\times}7H_2O$ (1 mg per 10 ml of water), 1 ml vitamin K3 ($1\,mg\,ml^{-1}$ in absolute ethanol), 1 ml $CaCl_2$ (0.8% w/v), 250 μl vitamin B12 solution ($0.02\,mg\,ml^{-1}$) and 0.5 g L-cysteine HCl.

For lipid analysis of non-BT strains: *S. paucimobilis* (ATCC 29837) was grown aerobically at 30 °C in nutrient broth (per litre: 5.0 g peptone and 3.0 g meat extract; pH 7.0). *B. fragilis* (DSM 2151), *Porphyromonas gingivalis* (DSM 20709), *B. uniformis* (DSM 6597), *B. vulgatus* H5_1 (DSM 108228), *B. vulgatus* (DSM 1447), *Prevotella veroralis* (ATCC 33779), *Phoecicola dorei* (DSM 17855), *Prevotella nigrescens* (DMS 13386) and *Prevotella copri* (DSM 18205) were grown anaerobically at 37 °C in BHIS. *Prevotella micans* (DSM 21469) was grown in anaerobic modified PYG medium and DSMZ medium 104 at 37 °C. *F. major* (DSM 103) was grown at 26 °C in DSM medium 7 (per litre: 1.0 g glucose, 1.0 g peptone and 1.0 g yeast extract; pH 7.0). *Acetobacter malorum* (DSM 14337) was grown at 28 °C in DSM medium 360 (per litre: 5.0 g yeast extract, 3.0 g peptone and 25.0 g mannitol). *N. acidiphilum* (DSM 19966) was grown at 28 °C in DSM medium 1199 (per litre: 1.0 g glucose, 1.0 g yeast extract and 1.0 g peptone; pH 5.5).

**Generation of BT knockouts and inducible SPT strain.** Genetic manipulations in the *B. thetaiotaomicron* VPI-5482 Δ*tdk* ('WT') strain were performed using double recombination from a suicide plasmid as previously described[21]. The generation of the *BT_0870* (*SPT*) knockout has been previously described[51]. To create the inducible SPT (iSPT) strain, three TetR cassettes were inserted into the Δ*BT_0870* genome with the constitutive PBT1311 promoter as previously described[28], with the native SPT (*BT_0870)* sequence reintroduced under the inducible P1TDP promoter. Induction of SPT was performed using anhydrotetracycline (aTC) in sterile ethanol in a range from 0 to $100\,ng\,ml^{-1}$. *BT_1522*, *BT_1523*, *BT_1524* *BT_1525* and *BT_1526* were knocked out using the same process in both the WT and iSPT strains. Complements for each gene were created in the iSPT knockout strains (excluding *BT_1524*), similarly using the constitutive PBT1311 promoter and native BT sequences, and integrated genomically just upstream of the inositol gene cluster. Plasmids, strains, primers and gene sequences are listed in Supplementary Table 1. All constructs were verified by Sanger sequencing. Sequencing of the full gene cluster in multiple recombinant strains revealed no mutations to explain the failure of the iSPTΔ*BT_1526*::*BT_1526* complement to produce inositol lipids.

**Bacterial lipid extraction and TLC.** BT strains were grown 14–20 h in BHIS; all other strains were grown in the media and temperatures described above to density. For 'standard' (non-acidic) lipid extraction: bacteria were pelleted at $3,500{\times}g$ for 15 min, the pellet washed in PBS and re-spun. The washed lipid pellets were lipid extracted by the Folch method[56], the organic fraction dried under nitrogen and the lipid film resuspended in 2:1 (v/v) chloroform:methanol. For PIP-DAG lipid extraction: to detect PIP-DAG, lipids were extracted according to the PI(3)P Mass ELISA kit (Echelon Biosciences) protocol. Cells from 50 ml BHIS cultures were pelleted at $3,500{\times}g$ for 15 min at 4 °C. Resuspended in 5 ml cold 0.5 M trichloroacetic acid (TCA), incubated 5 min on ice and pelleted at $3,500{\times}g$ for 15 min at 4 °C. The pellets were washed twice in 3 ml 5% TCA with 1 mM EDTA, then neutral lipids were extracted twice by vortexing the pellet in 3 ml 2:1 methanol:chloroform for 10 min. The resulting pellets were extracted into 2.25 ml methanol:chloroform:12 N HCl (80:40:1) and 0.75 ml chloroform, combined with 1.35 ml 0.1 N HCl and vortexed. The lower fraction was dried under nitrogen and resuspended in 20:9:1 chloroform:methanol:water for TLC.

*TLC of lipids.* Lipid extracts were applied to a silica HPTLC plate with concentration zone (Supelco 60768), with loading volumes normalized to the optical density $(OD)_{600}$ of original cultures. Plates were developed in a 62:25:4 (v/v) chloroform:methanol:ammonium hydroxide (25% $NH_3$ basis) system (for standard lipid extractions) or 48:40:7:5 chloroform:methanol:water:ammonium hydroxide (for PIP-DAG extractions), then sprayed with primuline ($0.1\,mg\,ml^{-1}$ in 4:1 v/v acetone:distilled $H_2O$), and imaged under ultraviolet transillumination (365 nm). TLC densitometry of scanned plates was used to measure the proportion of SLs using Fiji[57]. Lipid standards include 16:0 phosphatidylinositol (Avanti 850141), 18:1 PI(3)P (Avanti 850150), ceramide phosphorylethanolamine (Sigma-Aldrich C4987), egg yolk phosphatidylethanolamine (Pharmacoepia), d18:1/18:0 ceramide (Cayman 19556) and d18:0 sphinganine (Avanti 860498).

**Sample preparation for HPLC–MS lipid analysis.** Washed bacterial pellets were frozen over liquid nitrogen and lyophilised to dryness. HPLC-grade methanol (1 ml) was added to the dried material and the mixture was sonicated for 3 min (on/off pulse cycles of 2 s on, 2 s off at 100% power) using a Qsonica ultrasonic processor (Model Q700) with a water bath cup horn adaptor (Model 431C2) and

water bath flow to maintain approximately room temperature. Samples were then moved to an end-over-end rotator and extractions proceeded for 12 h. Samples were then centrifuged at $18,000{\times}g$ for 30 min at 4 °C. The supernatant was transferred to a fresh centrifuge tube and solvent was dried with a ThermoFisher Savant SpeedVac SPD130DLX. The dried material was resuspended in 200 μl HPLC-grade methanol, briefly sonicated and centrifuged as before. The concentrated extract was transferred to an HPLC vial with a 300 μl glass insert and stored at 4 °C until further analysis.

**HPLC–MS instrumentation.** LC–MS analysis was performed on a ThermoFisher Vanquish Horizon UHPLC System coupled with a ThermoFisher TSQ Quantis Triple Quadrupole mass spectrometer equipped with a HESI ion source. All solvents and reagents for HPLC–MS were purchased as Optima LC–MS grade (ThermoFisher).

**HPLC–MS/MS generalized method.** This method was used for the analysis of glycerophospholipids and sphingolipids, excluding PIP-DAG. Mobile phase A was 94.9% water, 5% methanol and 0.1% formic acid (v/v) with 10 mM ammonium acetate. Mobile phase B was 99.9% methanol and 0.1% formic acid (v/v). Extract (1 μl) was injected and separated on a mobile phase gradient with an Agilent Technologies InfinityLab Poroshell 120 EC-C18 column (50 mm × 2.1 mm, particle size 2.7 μm, part number: 699775-902) maintained at 50 °C. A/B gradient started at 15% B for 1 min after injection, increased linearly to 100% B at 22 min and held at 100% B for 5 min, using a flow rate $0.6\,ml\,min^{-1}$. Full Scan Q1 mass spectrometer parameters: spray voltage 2.0 kV for negative mode, ion transfer tube temperature 350 °C, vaporizer temperature 350 °C; sheath, auxiliary and spare gas 60, 15 and 2 psi, respectively. Tandem mass spectrum analysis was carried out with Product Ion Scan mode with the following additions: collision energy 30 V and CID gas 1.5 mTorr.

**HPLC–MS/MS method for phosphatidylinositol phosphates.** The method was slightly modified from Bui et al.[58]. Mobile phase A was 99.9% water and 0.1% *N,N*-diisopropylethylamine (v/v) with 10 μM disodium EDTA. Mobile phase B was 99.9% acetonitrile and 0.1% *N,N*-diisopropylethylamine (v/v). Extract (3 μl) was injected and separated on a mobile phase gradient with a Kinetex EVO C18 UHPLC column (2.1 × 150 mm, 1.7 μm) (Phenomenex, PN:00F-4726-AN) maintained at 60 °C. A/B gradient started at 38% B for 6 min after injection, increased linearly to 100% B at 12 min and held at 100% B for 3 min, using a flow rate $0.35\,ml\,min^{-1}$. Full Scan Q1 mass spectrometer parameters: spray voltage 4.5 kV for negative mode, ion transfer tube temperature 325 °C, vaporizer temperature 350 °C; sheath, auxiliary and spare gas 50, 15 and 1 psi, respectively. Tandem mass spectrum analysis was carried out with Product Ion Scan mode with the following additions: collision energy 30 V and CID gas 1.5 mTorr.

**LC–MS/MS analysis of PIP-DAG lipids for acyl chain determination.** To identify the acyl chain distribution in PIP-DAG lipids, WT and Δ*BT_1526* strains were grown and lipid extracted (using the PIP-DAG-specific acidic extraction) as described above. A Dionex UltiMate 3000 HPLC system (ThermoFisher) coupled with a high-resolution mass spectrometer with an electrospray ionization source (Impact II mass spectrometer, Bruker) was used for the analysis of the lipid extracts. The separations were performed on a hydrophilic interaction liquid chromatography column (L × inner diameter 150 cm × 3 mm, 2.6 μm particle size) held at a constant temperature of 60 °C. The mobile phase consisted of solvent A (0.1% formic acid and 2 mM ammonium formate in water:acetonitrile 40:60 (v:v)) and solvent B (0.1% formic acid and 2 mM ammonium formate in methanol). The following gradient: A/B 35/65 (0 min), 35/65 (from 2nd min), 60/40 (from 4th min), 60/40 (from 6th min); 35/65 (from 8th min) and 25/75 (at 10th min) was applied for elution with a constant flow rate of $0.6\,ml\,min^{-1}$. The injection volume was 45 μl. The operating parameters of the mass spectrometer were as follows: the spray needle voltage was set at 3.5 kV, nitrogen was used both as the nebulizing gas (1.5 bar) and the drying gas ($5\,l\,min^{-1}$), and the drying temperature was 200 °C. Data were acquired in data-dependent acquisition mode, with the 3 most intensive parent ions chosen for MS/MS acquisition. The scanning range was 50–1,500 *m/z* and the scanning rate was 2 Hz in the negative ion mode. The collision energy was 65 eV, with nitrogen used as collision gas. Full MS chromatogram was used for the quantification purpose (Skyline software version 21.1), while the MS/MS spectrum was used for identification and/or confirmation of PIP-glycerolipid (Compass Data Analysis version 4.3).

**Fatty acid methyl ester (FAME) analysis of lipid acyl chains by GC–MS.** Bacterial cultures were inoculated from overnight stationary cultures 1:500 (v/v) into 100 ml BHIS media and grown for 16 h at 37 °C, to an $OD_{600}$ of 0.30–0.45. Cultures were centrifuged for 40 min at $3,500{\times}g$ and the pellet was washed twice with PBS before Folch extraction in 10 ml 2:1 chloroform:methanol. The organic fraction was reduced by rotary evaporation (60 °C, 500 mbar vacuum) and dried under nitrogen to obtain the final lipid mass. All lipids were lifted in 200 μl 2:1 chloroform:methanol and used for FAME analysis as described below.

Determination of fatty acid composition by FAME derivatization was performed as previously described[59]. Each lipid extract (80 μl) was dried under nitrogen, 2 ml of MeOH with 2% $H_2SO_4$ and 800 μl toluene were added, and the samples were heated for 10 min at 100 °C. After cooling, 4 ml of nanopure water

and 800 μl hexane were added, followed by vortexing. After centrifugation for 5 min at 2,000×$g$, the upper layer was removed and dried under nitrogen, then lifted in 500 μl hexane and filtered (PVDF membrane) for GC–MS analysis.

Samples were run on an Agilent 7890B gas chromatograph outfitted with a Supelco OmegaWax 250 column (30 m × 250 μm × 0.25 μm) coupled to single quadrupole mass spectrometer 5977B. A sample (1 μl) was injected in splitless mode (pressure 10.5 psi, septum purge flow 0.5 ml min⁻¹, purge flow to split vent 50 ml min⁻¹ at 0.75 min) with helium (1 ml min⁻¹) as the carrier gas. The temperature cycle was as follows: 2 min at 100 °C, followed by a 10 °C min⁻¹ ramp to 1 min at 140 °C, followed by a 5 °C min⁻¹ ramp to a final 2 min hold at 270 °C. The detector was at 250 °C and the ionization source at 225 °C. Full scan detection was run in positive mode on an *m/z* interval of 50–650. Peak quantification was performed using external calibration curves with the corresponding FAME in concentrations from 3 to 250 μg ml⁻¹ (with the KEL-FIM-FAME-5 mixture 4210 from Matreya and additional individual FAME standards from Cymit Quimica), and lipid abundance reported as the molar percentage of total acyl chains in each culture. Data acquisition and quantitation were performed with Agilent MassHunter software.

**Fractionation of PI-DAG and PI-DHC lipids.** To purify PI-DAG and PI-DHC lipid fractions, 200 ml of WT BT was grown for 16 h in BHIS, washed and Folch-lipid extracted. The entire lipid extract was applied to a PLC plate (Merck silica gel 60 F₂₅₄ 0.5 mm with concentrating zone) and separated in a 60:30:2:4 chloroform:methanol:H₂O:ammonium hydroxide solvent system. To identify lipid bands, the side of the plate was sprayed with primuline and ultraviolet imaged. Horizontal sections were scraped and the silica (with lipid) was re-extracted in 5 ml of 2:1 chloroform:methanol, then poured through a fritted column to remove the silica. The lipids (in solvent) were dried under nitrogen and the lipid film was lifted in 2:1 chloroform:methanol.

Lipid fractions were first screened by TLC in a 60:30:4 chloroform:methanol:ammonium hydroxide system, visualized with primuline and ultraviolet imaged. Each lipid fraction was divided: one-half was processed for FAME analysis as described above, the remaining half was analysed by MALDI mass spectrometry (as described under 'inositol uptake').

**Purification and enzymatic characterization of BT_1526.** The synthetic gene encoding the BT_1526 open reading frame (wild type) was ordered from Genscript and cloned into a pET-28a expression plasmid with a six-histidine tag at the N terminus. The pET-BT_1526 plasmid was used to transform *E. coli* BL21 (DE3) cells for overexpression. The BT_1526 MIPS protein was expressed by culturing the transformed cells in LB medium supplemented with 35 μg ml⁻¹ kanamycin at 37 °C and 200 r.p.m. shaking until the cells reached the mid-exponential growth stage (OD₆₀₀ = 0.5). Protein expression was then induced by the addition of 0.1 mM isopropyl β-d-1-thiogalactopyranoside for 5 h with a reduced temperature of 30 °C and shaking at 200 r.p.m. Cells were collected by centrifugation at 4,000×$g$ at 4 °C and sonicated in 10x v/w HisA buffer (50 mM Tris-HCl, 20 mM NH₄Cl, 0.2 mM dithiothreitol and 30 mM imidazole, pH 7.5) to lyse the cells. The lysate was clarified by centrifugation at 35,000×$g$ at 4 °C and the supernatant was loaded onto a 5 ml HisTrap column (Cytiva) equilibrated with HisA buffer. Unbound proteins were washed off the column with 20 column volumes of HisA before elution of the tagged BT_1526 with HisB buffer (20 mM NH₄Cl, 0.2 mM dithiothreitol and 500 mM imidazole, pH 7.5). The protein was then subjected to size-exclusion chromatography for polishing and buffer exchange. A Superdex S200 column was equilibrated with GF buffer (50 mM Tris-HCl and 150 mM NaCl) and the sample added for isocratic elution over 1.2 column volumes. Fractions from size-exclusion chromatography were analysed by 15% SDS–PAGE and fractions containing the pure BT_1526 protein were pooled and used in subsequent experiments. The yield of recombinant BT_1526 was typically >20 mg l⁻¹ of *E. coli* culture.

**Mass spectrometry analysis of purified MIPS.** Samples were analysed in the positive ion mode using HPLC coupled to a Waters Synapt G2 QTOF with an electrospray ionization source (ESI). About 5–10 μl 10 μM protein was injected onto a Phenomenex C4 3.6 μ column. The conditions for the QTOF are as follows: source temperature 120 °C, back pressure 2 mbar and sampling cone voltage 54 V. The protein was eluted with a 12 min gradient, starting at 5% acetonitrile with 0.1% formic acid to 95% acetonitrile. The resulting spectra were processed and the charge state distributions deconvoluted using MassLynx V4.1 software.

**Assay of BT_1526 for MIPS activity.** The purified BT_1526 was assayed for MIPS activity on the basis of a method published by Barnett[60] (Fig. 3a) with $n = 4$ per substrate concentration tested. The assay was as follows: 1 μM enzyme, 0–50 mM D-glucose-6-phosphate and 0.8 mM NAD+ for 1 h at 25 °C. The reaction was quenched with 20% TCA, then 0.2 M NaIO₄ was added for 1 h at room temperature. Then 1.5 M Na₂SO₃ was added to remove excess NaIO₄. The reagent mix was incubated for 1 h at room temperature, then the absorbance was measured on a BioTek Synergy HT plate reader at 820 nm. Values were determined with reference to inorganic phosphate standards.

**Crystallization of BT_1526.** MIPS was initially screened using commercial kits (Molecular Dimensions and Hampton Research). The protein concentration

was 11.4 mg ml⁻¹. The drops, composed of 0.1 μl or 0.2 μl of protein solution plus 0.1 μl of reservoir solution, were set up using a Mosquito crystallization robot (SPT Labtech) using the sitting drop vapour diffusion method. The plates were incubated at 20 °C and the initial hits were suitable for diffraction experiments. The condition yielding crystals that were subjected to X-ray diffraction was PACT F6 (Molecular Dimensions, 200 mM sodium formate 100 mM bis tris propane pH 6.5 and 20% (w/v) PEG 3350). The sample was cryoprotected with the addition of 20% PEG 400 to the reservoir solution.

**Data collection, structure solution, model building, refinement and validation of BT_1526.** Diffraction data were collected at the synchrotron beamline I04 of Diamond light source at a temperature of 100 K. The data set was integrated with XIA2[61] using DIALS[62] and scaled with Aimless. The space group was confirmed with Pointless[63]. The phase problem was solved by molecular replacement with Phaser[64], using PDB file 3QVT as search model. The model was refined with refmac[65] and manual model building was done with COOT[66]. The model was validated using Coot and Molprobity[67]. Other software used were from CCP4 cloud and the CCP4 suite[68]. Figures were made with ChimeraX[69].

**Inositol uptake.** To determine whether BT is capable of taking up exogenous inositol and incorporating this into lipids in the absence of MIPS/BT_1526, we prepared BMM containing solely glucose as a carbon source or an equivalent molar mass of 1:1 myo-inositol:glucose. WT and ΔBT_1526 strains were inoculated into each medium from dense overnight cultures ($n = 2$) and grown for 16 h at 37 °C. Lipids were extracted using the Folch method ('standard' (non-acidic) lipid extraction, described above) and lipids were measured by MALDI MS.

For MALDI MS analysis, lipid extracts (100 μg ml⁻¹ in MeOH) were mixed 1:1 (v/v) with 1,5-diaminonapthalene (10 mg ml⁻¹ in acetonitrile/water/trifluoroacetic acid 7:3:0.01 (v/v/v)) and vortexed. The resulting solution (1 μl) was spotted onto the MTP 384 ground steel target and allowed to dry at room temperature. Measurements were performed using a Bruker scimaX 7T 2xR FTICR mass spectrometer and ftmsControl V2.3.0 (Bruker Daltonics). Before the measurement, external quadratic calibration was performed with sodium formate (5 mM in 2-propanol/water 90:10 (v/v)) injected at a flow rate of 120 μl h⁻¹ (nebulizer set to 1.5 bar with a dry gas flow and temperature of 4 l min⁻¹ and 200 °C, respectively) using the ESI source.

All data were acquired in negative ion mode using the MALDI ionization source. Laser intensity was set to 12%. Per spot, 50 laser shots and a frequency of 104 Hz was used. Spectra were acquired with small laser focus and a smart walk of 1,000 μm width (grid increment set to 9, offset set to 1). Ions were detected in the *m/z* range of 110–2,000, with Q1 set to *m/z* 800. For ion transfer, voltages of funnel 1 and skimmer 1 were set to −150 V and −15 V, respectively, with the funnel radio frequency amplitude adjusted to 70 Vpp. Octopole frequency and radio frequency amplitude were set to 5 MHz and 350 Vpp, respectively. The frequency of the transfer optics was set to 4 MHz and a time of flight of 1 ms was used. Mass accuracy was ensured using the prominent lipid peak at *m/z* 662.476082 ((PE30:0 − H)⁻, $C_{35}H_{69}O_8NP$) as a reference mass for single-point calibration. The resulting spectra were processed with Bruker Compass DataAnalysis V5.2 software (Bruker Daltonik). Average intensities for 8 spots were calculated for each sample.

**Growth curve analysis and LL-37 resistance.** All growth curves were performed in 96-well format in an anaerobic chamber (95% N₂, 5% CO₂ atmosphere). Plates were incubated at 37 °C and OD₆₀₀ measured every 8 min for 24 h in a Victor Nivo plate reader (PerkinElmer). For growth curves: to determine alterations to growth dynamics in the knockout strains of the inositol lipid cluster, dense overnight cultures of each BT strain in BHIS were centrifuged at 16,000×$g$ for 1 min, the supernatant removed and the cells resuspended in an OD-normalized volume of BHIS or BMM before aliquotting 200 μl per well in a sterile 96-well plate ($n = 4$). For AMP resistance: to identify changes of each strain's sensitivity to the AMP LL-37, dense overnight cultures of each BT strain were OD-normalized and diluted to a starting OD₆₀₀ of ~0.15 in a total of 200 μl BMM per well, each well supplemented with 0–64.0 μg ml⁻¹ LL-37 (InvivoGen) in duplicate per strain and AMP concentration. Inhibition due to LL-37 was graphically modelled by percent OD at 0 ng μl⁻¹ LL-37: (maximum OD in 24 h − minimum OD in 24 h)/ avg(maximum − minimum OD at 0 ng ml⁻¹ per strain) × 100. $IC_{50}$ was calculated in Prism by fitting 'concentration inhibitor vs normalized response with variable slope' and between-strain comparisons using Tukey's multiple comparisons. The data shown are representative of two experiments.

**RNA-seq of BT at varied levels of SPT induction.** Overnight cultures were used to inoculate (in duplicate) BMM media 1:2,500 (v/v) uninduced or at 1 of 5 varied aTC concentrations (0, 0.2, 1.0, 5.0, 100 ng ml⁻¹), and incubated at 37 °C for 15 h to an OD₆₀₀ of 0.10–0.17. Cultures were spun at 3,500×$g$ for 15 min and RNA was extracted from the bacterial pellet with QIAzol lysis reagent and the miRNeasy mini kit (Qiagen). Ribosomal RNA was removed with the Bacterial RiboMinus transcriptome isolation kit (Invitrogen) and the library prepared with the TruSeq stranded total RNA library kit (Illumina); libraries were pooled at 9 per lane and sequenced by HiSeq3000 (Illumina).

Quality assessment of reads was performed using FastQC pre- and post-quality filtering with bbduk (quality cut-off 20)[70]. Reads were aligned to the Ensembl *B.*

*thetaiotaomicron* VPI-5482 genome with bowtie2 and assigned using htseq-count (alignment quality cut-off 10)[71–73]. Differential expression analysis was performed with EdgeR and limma:[74,75] reads assigned to rRNA genes, 'ambiguous' or 'no feature' were removed, lowly expressed genes were filtered and gene expression distributions were normalized (method Trimmed Means of M values, 'TMM'). Count data from samples, which were in duplicate, were further normalized by Bayes moderated variance before calculation of differential expression (adjusted *P* value via Benjamini-Hochberg method). Annotations were assigned from the JGI IMG database. Pathway enrichment was performed using PANTHER[76]. Heat maps were generated with pheatmap using normalized log₂ expression values, scaled by row with Euclidean clustering. Gene expression change cut-off for differential expression (Supplementary Tables 3 and 4) was $\log_2$ fold change >1.5.

**Capsule analysis.** For capsule extraction, 800 ml BHIS cultures of each strain were grown for 18 h at 37 °C. Bacteria were pelleted at 5,000 × *g* for 40 min at 4 °C and the pellet washed twice in PBS. The pellet was resuspended in 20 ml 1% phenol in double distilled H₂O and shaken 2 h at r.t., then centrifuged at 5,000 × *g* for 45 min at 4 °C. The upper layer was ethanol precipitated with 4x volume ethanol at −20 °C overnight, then centrifuged at 5,000 × *g* for 30 min at 4 °C, washed in 80% ethanol and allowed to dry. The pellet was resuspended in 20 mM Tris-HCl (pH 7.4), 2.5 mM MgCl₂ and 0.5 mM CaCl₂. Benzonase nuclease (Millipore) was added to a final concentration of 75 U ml⁻¹, followed by incubation with gentle shaking at 37 °C for 20 h. Proteinase K (Roth) was added to a final concentration of 60 ug ml⁻¹, followed by incubation with gentle shaking at 60 °C for 20 h. Samples were dialysed exhaustively against distilled water (20 kD Slide-A-Lyzer; ThermoFisher) and dried by SpeedVac (ThermoFisher) at 40 °C before analysis.

Glycosyl composition analysis of the extracted capsule samples was performed by the University of Georgia Complex Carbohydrate Research Center, using combined GC–MS of the *O*-trimethylsilyl (TMS) derivatives produced from the sample by acidic methanolysis; these procedures were carried out as previously described[77–79]. GC–MS analysis of the TMS methyl glycosides was performed on an AT 7890A GC interfaced to a 5975B MSD, using an EC-1 fused silica capillary column (30 m × 0.25 mm inner diameter).

**Electron microscopy.** Bacterial strains (iSPT and iSPTΔBT_1526) were grown in BMM at 0 and 100 ng ml⁻¹ induction of SPT. Imaging was performed by the Electron Microscopy Core at the Max Planck Institute for Biology Tübingen (Germany). For SEM, cells were fixed in 2.5% glutaraldehyde/4% formaldehyde in PBS for 2 h at room temperature and mounted on poly-L-lysine-coated cover slips. Cells were post-fixed with 1% osmium tetroxide for 45 min on ice. Subsequently, samples were dehydrated in a graded ethanol series, followed by critical point drying (Polaron) with CO₂. Finally, the cells were sputter-coated with a 3 nm thick layer of platinum (CCU-010, Safematic) and examined with a field emission scanning electron microscope (Regulus 8230, Hitachi High Technologies) at an accelerating voltage of 3 kV.

**Mouse experiments.** The mouse experiment was performed in accordance with the German legislation on protection of animals, with permission to conduct the study obtained from the regional animal welfare committee of the Eberhard Karls Universität Tübingen, registration number EB 03/21 M. Female germ-free C57BL/6 mice aged 4–6 weeks were orally inoculated with WT, ΔBT_1522, ΔBT_1523, ΔBT_1526 or a WT:ΔBT_1523 strain combination (~10⁸ colony-forming units (c.f.u.) per mouse in PBS, *n* = 8). Mice were co-housed for 14 d with a 12 h light:dark cycle at 22 ± 2 °C and ~55 ± 10% humidity, and euthanized by decapitation. Caecal contents were snap-frozen in liquid nitrogen for further analyses.

*Colonization quantification (mono-associations).* To quantify total bacterial load in caecal contents, ~100–200 mg of caecal contents were fully homogenized in sterile water at a ratio of 411 µl per 100 mg caecal contents, then centrifuged at 500 × *g* for 2 min at 2 °C. The supernatant was diluted in sterile water and boiled at 100 °C for 10 min, then diluted to a final concentration of 1:2,500 as quantitative PCR DNA input. qPCR was performed using the CFX Connect Real-Time PCR detection system (Bio-Rad), KiCqStart SYBR Green qPCR ReadyMix (Sigma-Aldrich) and 300 nM primers specific for single-copy gene *BT_1521* (Fwd: AATGGTTCCCATCACCCACC, Rev: TACGTAACTCCCGTCGGACT; designed using Primer-BLAST, NCBI[80]) using the following cycling conditions: 95 °C 0:30, 39 cycles (95 °C 0:05, 60 °C 0:15, 72 °C 0:25), melt curve 55 °C to 95 °C, 4 °C hold. Each sample was tested in triplicate and all data points fell within the standard curve of WT BT DNA.

*Colonization quantitation (competition experiment).* To quantify the strain ratio in the caecal contents of mice inoculated with both WT and ΔBT_1523 BT strains, caecal contents were DNA extracted using the AllPrep PowerFecal DNA/RNA kit (Qiagen); the inoculum was DNA extracted using the Jena Bioscience Bacteria DNA Preparation-Solution kit. qPCR conditions were as for mono-association quantification, using primers specific to the 16S gene as a measure of total bacterial load (27F:AGAGTTTGATCCTGGCTCAG, 1391R: GACGGGCGGTGWGTRCA)[81,82], and primers specific to BT_1523 (Fwd: CAGGGCAACAAACGCAATGA, Rev: GATATGATGGACGGGCGTGT)

designed using Primer-BLAST (NCBI[80]). Abundance of the WT strain was determined as the ratio of BT DNA determined by BT_1523 primers (as the ΔBT_1523 strain has no amplification with this primer set) over the total quantity of BT DNA (measured by 16S primers).

**Phylogenies of homology to BT inositol lipid metabolic enzymes in diverse bacteria.** For the smaller phylogeny of diverse sphingolipid producers (Fig. 5d), homology to BT inositol and inositol lipid metabolism enzymes BT_1522, BT_1523, BT_1525 and BT_1526 in the indicated species was identified using NCBI BlastP[83]. For the larger phylogeny of Bacteroidetes and related genera (Fig. 6), all representative species for *Bacteroides, Prevotella, Parabacteroides, Porphyromonas, Flavobacterium, Sphingobacterium* and *Chlorobium* genera with nomenclature recognized in the List of Prokaryotic names with Standing in Nomenclature (LPSN)[84] were tested for homology to BT inositol metabolism enzymes BT_1522, BT_1523, BT_1525, BT_1526 and BT_4744. For phylogenetic comparison in both trees, 71 single-copy genes present in all genomes (HMM profile Bacteria_71) were identified and concatenated using Anvi'o[85], with alignment using MUSCLE[86]. RAxML[87] was used to generate a maximum-likelihood tree (Protcat substitution model, Dayhoff matrix, Hill-climbing algorithm, 50 bootstrap iterations). Strain accession numbers and BlastP results are in Supplementary Table 5.

**Statistics and reproducibility.** Statistical significance for caecal colonization and competition experiments, antimicrobial peptide (LL-37) resistance and capsule composition was calculated by one-way analysis of variance (ANOVA) with Tukey's multiple comparisons. Statistical analysis was performed with GraphPad Prism version 9.0 and statistical significance is indicated by: ***$P < 0.001$; **$P < 0.01$; *$P < 0.05$; NS (not significant) $P > 0.05$. In the RNA-seq experiment, *P* values were calculated from empirical Bayes moderated *t*-statistics and adjusted according to the Benjamini-Hochberg method; only significantly differentially expressed genes with >1.5 absolute log₂ fold change are reported. The '*n*' reported in each experiment represents an individual biological replicate for the relevant experiment (for example, mouse caecal sample, bacterial culture). No statistical methods were used to pre-determine sample sizes, but our sample sizes are similar to those reported in previous publications[51]. Data collection and analysis were not performed blind to the conditions of the experiments. No animals or data were excluded from the analyses. In mouse experiments, mice were randomly assigned to a treatment condition to randomize age and litter of origin. For in vitro experiments, following strain generation, identical treatments were performed (for example, lipid extraction and analysis, capsule extraction and analysis, AMP resistance and growth curves) so randomized allocation was not necessary. TLC experiments were repeated independently at least two times with similar results.

**Reporting summary.** Further information on research design is available in the Nature Research Reporting Summary linked to this article.

## Data availability

The BT_1526 MIPS structure analysed during the current study is available in the Protein Data Bank repository, PDB ID 7NWR. RNA-seq reads and data are available at NCBI GEO (https://www.ncbi.nlm.nih.gov/geo/) under accession number GSE193734. Mass spectrometry files and all unique strains generated in this study are available from the corresponding author upon request. All remaining data generated during this study are included in this Article and its Supplementary Information. Source data are provided with this paper.

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

## Acknowledgements

This work was supported by the Max Planck Society. We thank K. Hipp and J. Berger of the Electron Microscopy Core Facility at the Max Planck Institute for Biology Tübingen for their expert imaging of the bacterial capsules; A. Goodman (Yale School of Medicine, USA) for providing relevant strains of *B. thetaiotaomicron*; D. Warschawski for training on fatty acid methyl ester analysis; the Diamond Light Source (Oxfordshire, UK) for beamtime (proposal mx24948) and staff of beamline I04. D.J.C. and J.M.-W. acknowledge the funding provided by the Biotechnology and Biological Sciences Research Council (BBSRC, grants BB/V001620/1 and BB/V00168X/1). This work was supported in part by NIH grant R24GM137782-01 to Parastoo Azadi of the Complex Carbohydrate Research Center at the University of Georgia.

## Author contributions

S.L.H. and R.E.L. conceived and planned the experiments. P.T. purified MIPS and performed its kinetic assay with input from D.J.C. J.M.-W. and A.B. performed the structural studies of MIPS. H.H.L., C.M.B. and D.L.V. performed mass spectrometry with input from E.L.J. J.L.W. coordinated the mouse experiment. S.L.H. performed all other experiments (strain generation, bacterial culturing, lipid extractions, TLC, FAME analysis, RNA-seq, capsule extractions, AMP resistance assays, mouse analyses and bioinformatic/phylogenetic analyses), analysed the data and wrote the first draft of the manuscript. S.L.H. and R.E.L. prepared the final manuscript. All authors provided comments and gave approval for publication.

## Funding

## Competing interests

The authors declare no competing interests.

## Additional information

**Extended data** is available for this paper at https://doi.org/10.1038/s41564-022-01152-6.

**Correspondence and requests for materials** should be addressed to Ruth E. Ley.

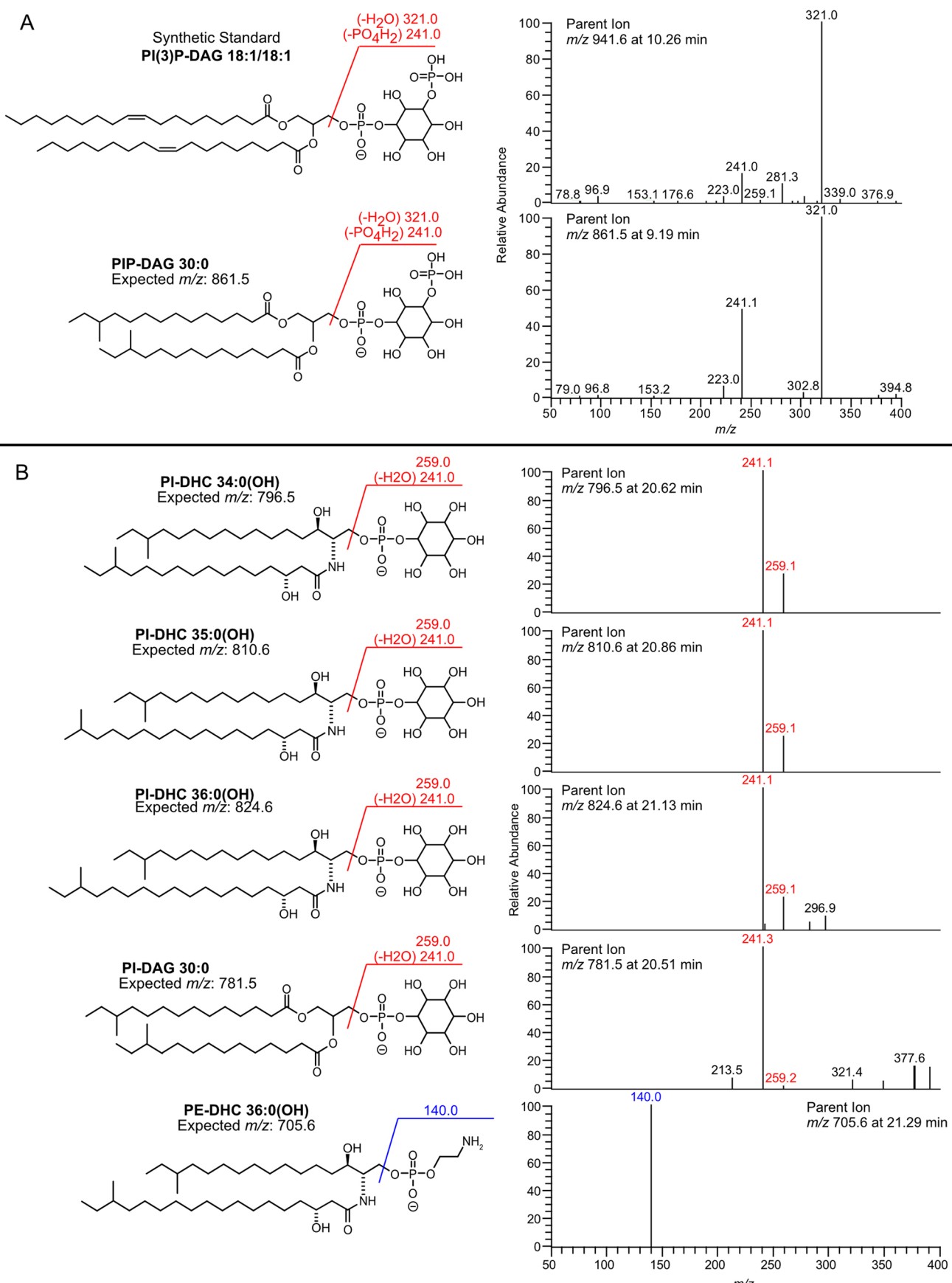

**Extended Data Fig. 1 | See next page for caption.**

**Extended Data Fig. 1 | Lipid structures and fragmentation patterns of BT inositol and ethanolamine lipids.** (A) Comparison of LC-MS/MS fragmentation patterns of ΔBT_1525 BT-derived phosphatidylinositol phosphate (PIP-DAG) with the synthetic standard, PI(3)P 18:1/18:1. (B) LC-MS/MS fragmentation patterns of lipid structures present in iSPT BT at 100 ng/mL aTC induction, including phosphoinositol dihydroceramide (PI-DHC) lipids (PI-DHC 34:0(OH), PI-DHC 35:0(OH), PI-DHC 36:0(OH)), PI-DAG 30:0, and PE-DHC 36:0(OH). Loss of the phosphoinositol head group is indicated at mass 259. Fragments characteristic for lipids with phosphoinositol-based headgroups are in red; those for phosphoethanolamine-based headgroups are in blue. Branching patterns of DHC-based lipids are predicted.

| | inos-P | PIP-DAG | PI-DAG | PI-DHC | PE-DHC | PE-DAG |
|---|---|---|---|---|---|---|
| WT | ✓ | ✓ | ✓ | ✓ | ✓ | ✓ |
| ΔBT_1522 | ✓ | ✓ | ✓ | X | ✓ | ✓ |
| ΔBT_1523 | ✓ | X | X | X | ✓ | ✓ |
| ΔBT_1524 | ✓ | ✓ | ✓ | ✓ | ✓ | ✓ |
| ΔBT_1525 | ✓ | ✓ | X | X | ✓ | ✓ |
| ΔBT_1526 | X | X | X | X | ✓ | ✓ |

**Extended Data Fig. 2 | Summary of lipid species and inositol metabolites produced by each WT or knockout strain.** Lipid color scheme is consistent with Fig. 2. Red x = not present; green check = present. Inositol-P ("inos-P") is described as present/absent dependent on the presence of the *BT_1526* gene. PIP-DAG = phosphatidylinositol phosphate; PI-DAG = phosphatidylinositol; PI-DHC = phosphoinositol dihydroceramide; PE-DHC = phosphoethanolamine dihydroceramide; PE-DAG = phosphatidylethanolamine. Yellow background for PIP-DAG presence/absence indicates PIP-DAG presence (determined by presence of downstream metabolites), but these lipids are not detectable by mass spectrometry, due to predicted fast turnover.

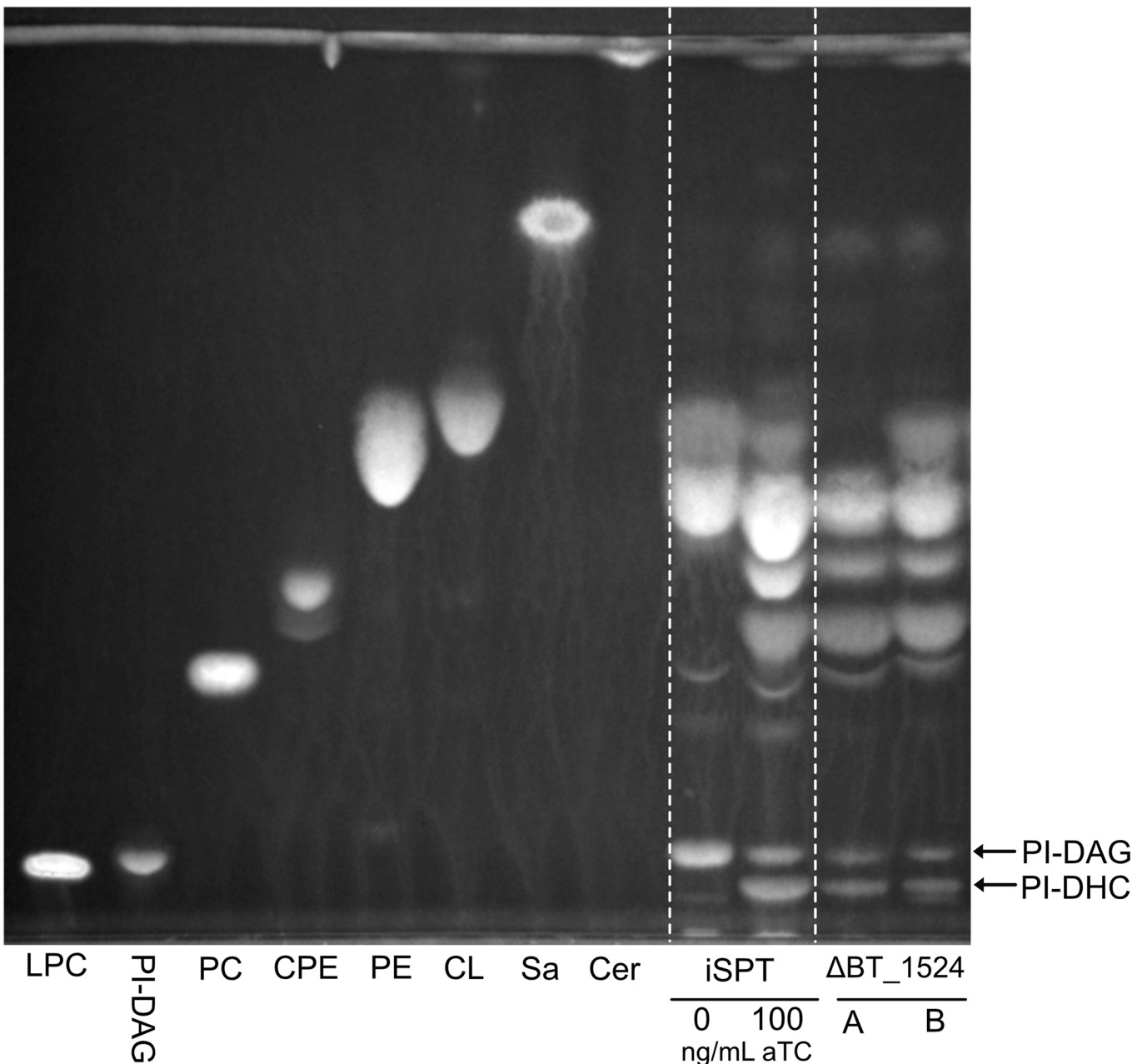

**Extended Data Fig. 3 | Lipid comparison of the ΔBT_1524 strain compared to iSPT.** TLC of lipid standards and Folch (non-acidic) lipid extractions from iSPT and ΔBT_1524 strains of BT. Lanes 1–8, left to right: LPC = 16:0 lyso-phosphatidylcholine; PI-DAG = 16:0 phosphatidylinositol; PC = 16:0 phosphatidylcholine; CPE = ceramide phosphoethanolamine; PE = egg yolk phosphatidylethanolamine; CL = cardiolipin (from bovine heart); Sa = d18:0 sphinganine; Cer = d18:1/18:0 ceramide. Following the dashed white line, in lanes 9–10, are iSPT BT lipid extracts from cells grown with 0 or 100 ng/mL aTC induction of SPT. In the final two lanes (11–12) are lipid extracts from ΔBT_1524 BT. "A" and "B" refer to independently generated knockout strains, both confirmed by Sanger sequencing. The retention factors for inositol lipids (phosphatidylinositol "PI-DAG" and phosphoinositol dihydroceramide "PI-DHC") are indicated by labels to the right of the figure.

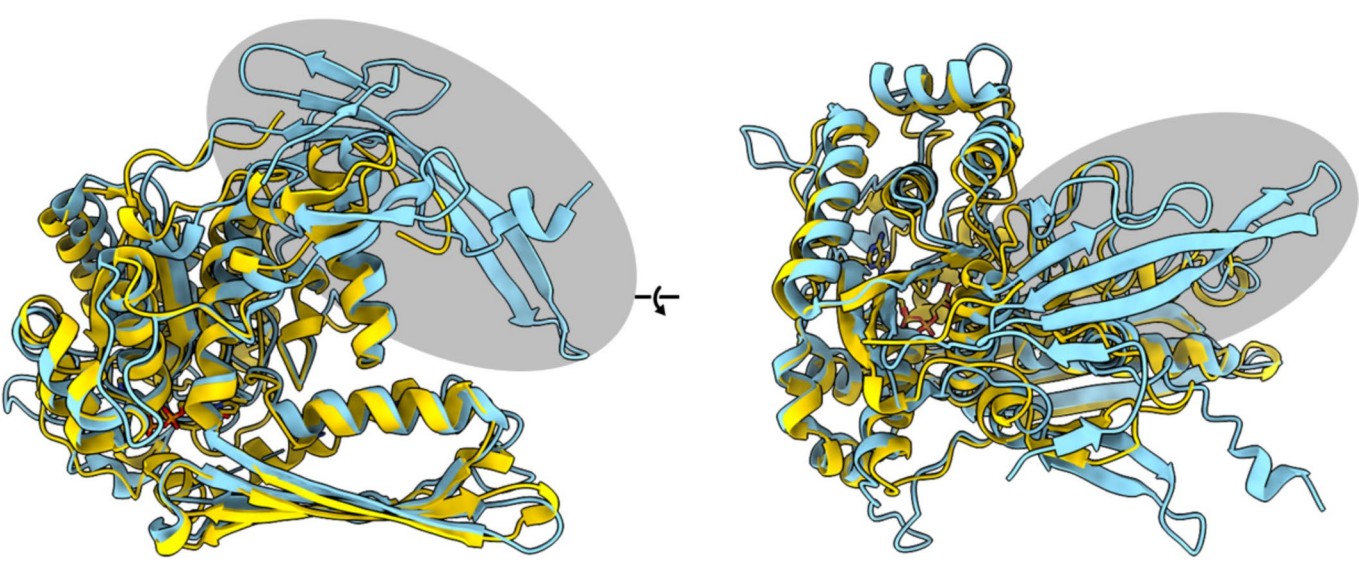

**Extended Data Fig. 4 | Structural comparison of prokaryotic and eukaryotic MIPS proteins.** Secondary structure alignment of MIPS BT_1526 (in yellow) and *Saccharomyces cerevisiae* MIPS (PDBID: 1P1i) (in cyan). The N-terminal extension present in eukaryotic MIPS structures is highlighted with a gray oval.

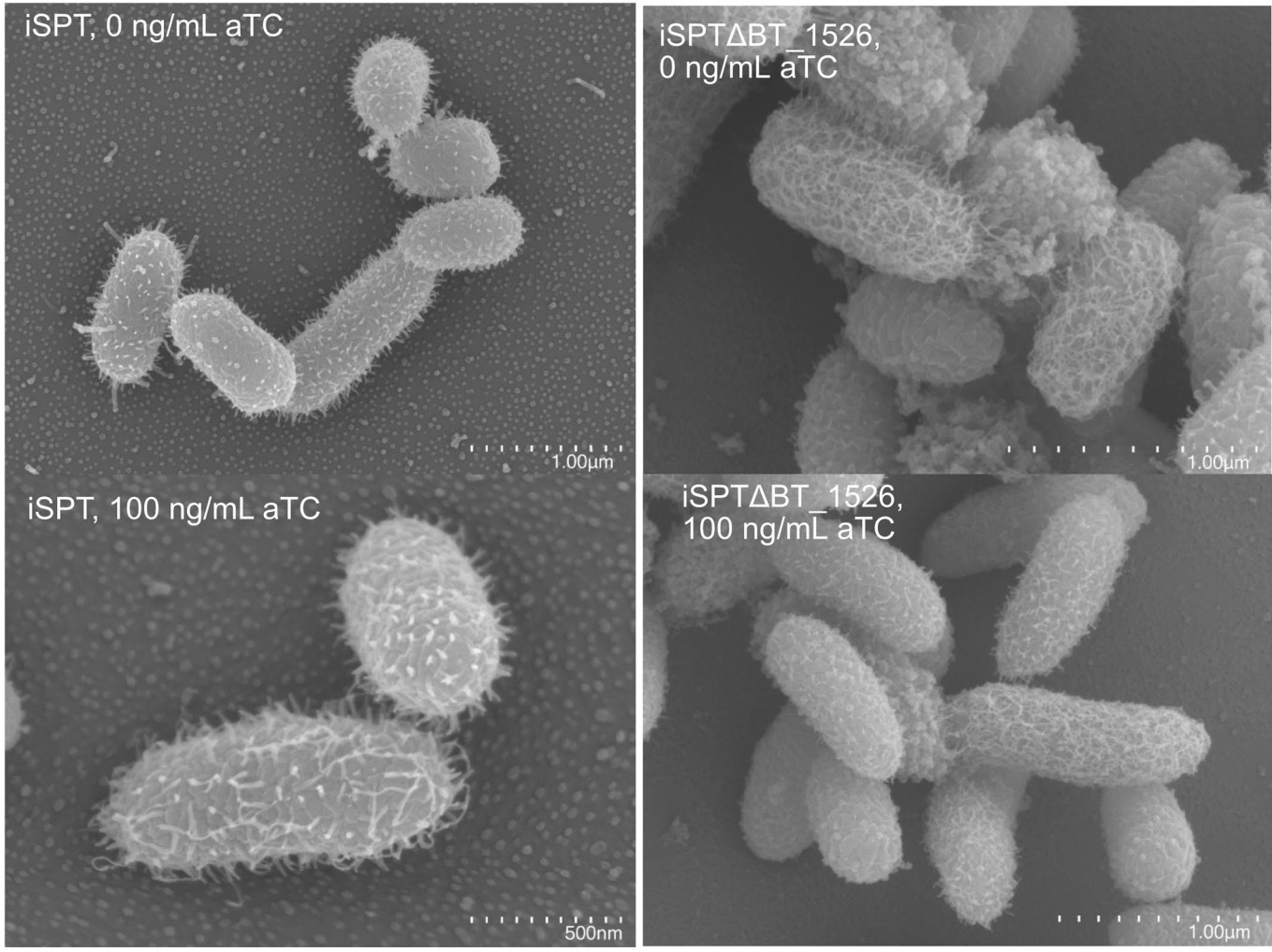

**Extended Data Fig. 5 | Scanning electron microscopy of the iSPT and ΔBT_1526 strains.** Cells were grown in minimal medium at 0 and 100 ng/mL anhydrotetracycline (aTC) induction of SPT prior to imaging by the Max Planck for Biology Tübingen Electron Microscopy Core Facility. Images are representative of multiple images derived from a single experiment.

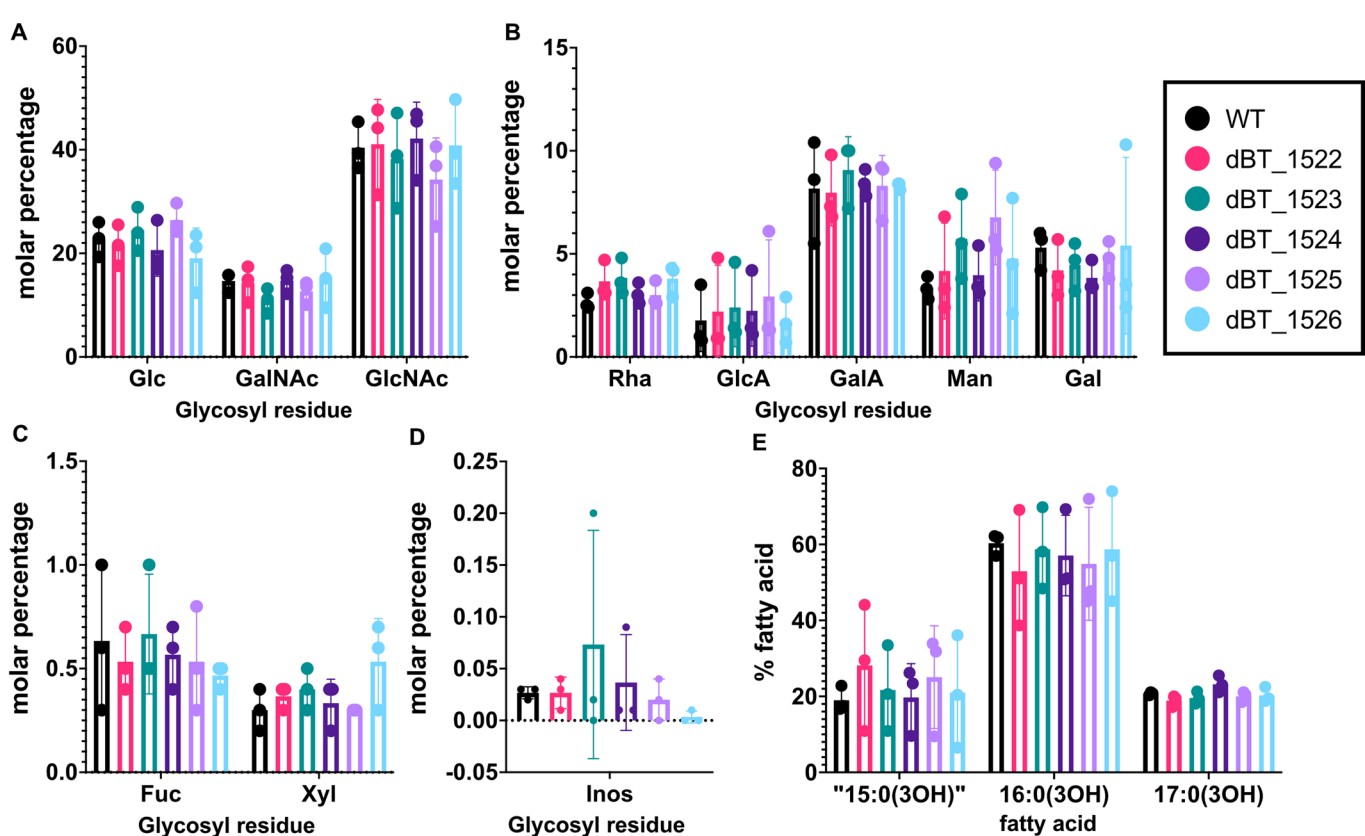

**Extended Data Fig. 6 | Capsular components of WT and inositol lipid knockout strains.** (A-D) Glycosyl residues detected in the capsular extraction from each strain, presented as percent molar abundance of total glycosyl residues detected (n = 3 biological replicates per strain tested; data represented as mean values ± S.D.), in separate figures to better compare relative abundances between residues at similar concentrations. Values are in Source Data Table 6. Glc = glucose; GalNAc = N-acetylgalactosamine; GlcNAc = N-acetylglucosamine; Rha = rhamnose; GlcA = glucuronic acid; GalA = galacturonic acid; Man = mannose; Gal = galactose; Fuc = fucose; Xyl = xylose; Inos = inositol. (E) Fatty acid species detected in the capsular extraction from each strain, presented as percent abundance of total fatty acids detected (n = 3 biological replicates per strain tested; data represented as mean values ± S.D.). "15:0(OH)" is in quotations due to uncertainty of its identity - this lipid had a diagnostic signal for a 3-OH fatty acid eluting at RT = 34.800 min, however we could not detect a diagnostic $m/z$ EI fragment (329) characteristic for hydroxypentadecanoic acid (15:0(3:OH)), instead finding an EI fragment at $m/z$ 319.

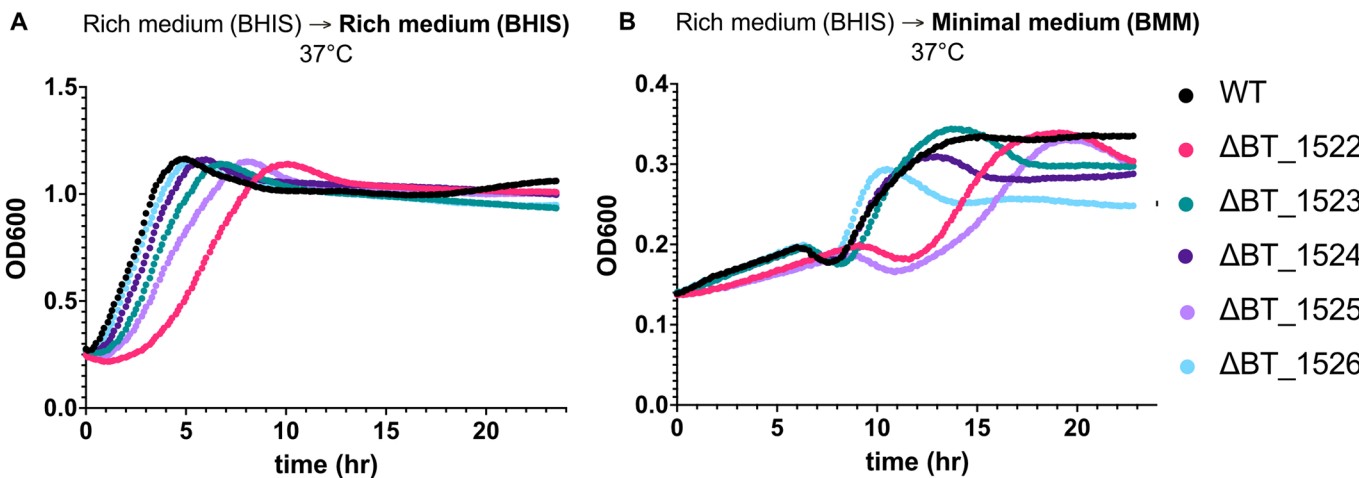

**Extended Data Fig. 7 | Growth curves of inositol lipid knockout strains in rich and minimal media.** (A) Growth curves of WT BT and inositol lipid knockout strains in rich medium (BHIS) at 37 °C in anaerobic conditions (all curves with average of n = 3 shown), measured in 96-well format as optical density at 600 nm (OD600). (B) Strain growth in minimal medium (BMM), inoculated using cultures previously growing in rich medium (BHIS).

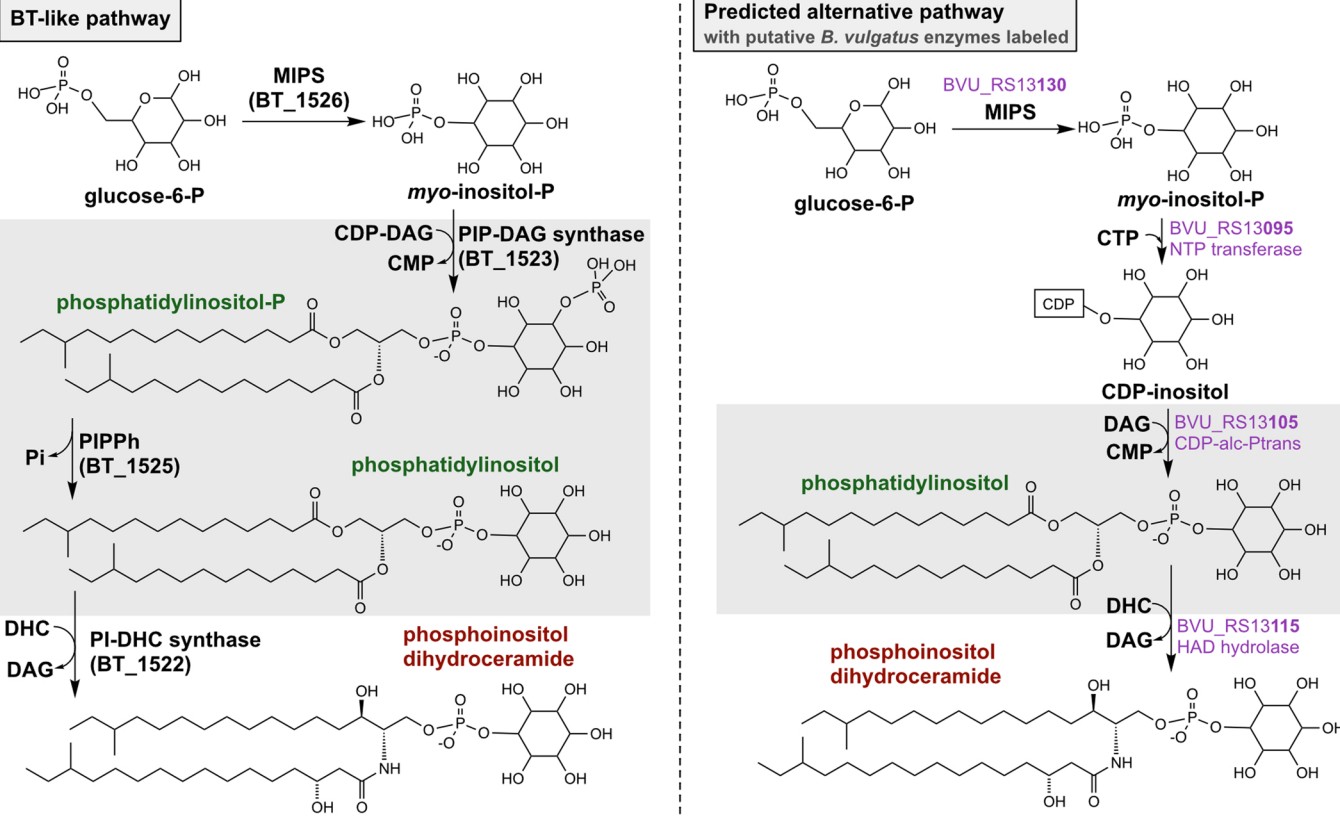

**Extended Data Fig. 8 | Pathway comparison between the BT inositol lipid cluster and the predicted alternative inositol lipid pathway.** At left, the BT-like inositol lipid synthesis pathway defined in this work. At right, the putative inositol lipid synthesis pathway genomically predicted for multiple Bacteroidetes spp., with predicted enzymes (in purple) from the *Bacteroides vulgatus* genome (NCBI reference sequence NC_009614.1). Inositol glycerophospholipids are on a gray background with green text; inositol sphingolipids are on a white background with red text. Branching patterns of BT DHC-based lipids and *B. vulgatus* lipids are predicted and not confirmed. SPT = serine palmitoyltransferase; MIPS = *myo*-inositol phosphate synthase; PIPPh = phosphatidylinositol phosphate phosphatase.

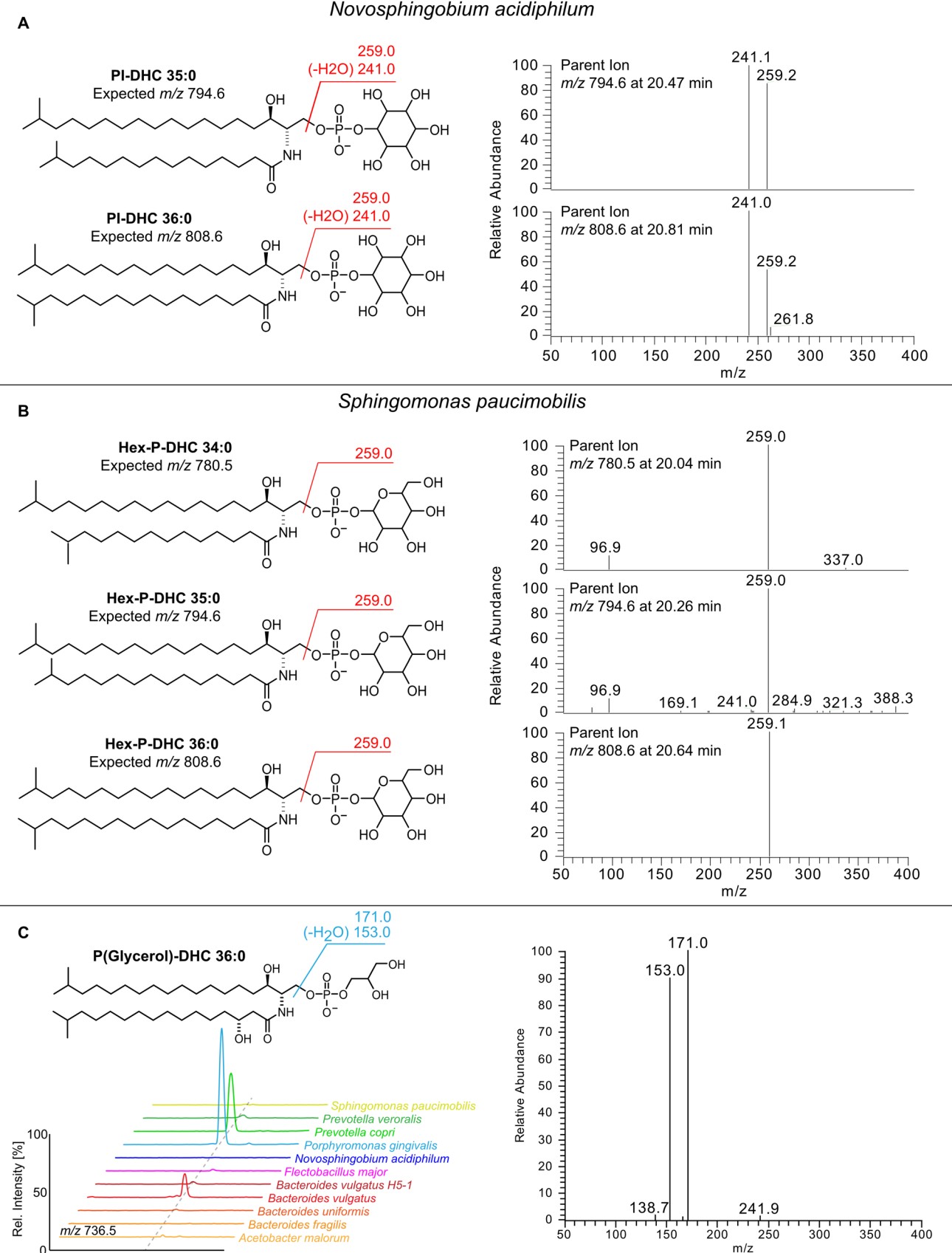

**Extended Data Fig. 9 | See next page for caption.**

**Extended Data Fig. 9 | Non-hydroxylated inositol-like lipid structures in diverse sphingolipid-producing species.** (A) LC-MS/MS fragmentation patterns of lipids extracted from *Novosphingobium acidiphilum* consistent with the synthesis of PI-DHC 35:0 and 36:0. (B) LC-MS/MS fragmentation pattern of lipids extracted from *Sphingomonas paucimobilis*, demonstrating the presence of a headgroup with the same mass as inositol phosphate (259) but lacking the characteristic fragment of this group (241). As such, the headgroup identity remains tentative and is represented with a phosphohexose. (C) LC-MS/MS spectra and fragmentation pattern of a P(Glycerol)-DHC 36:0 structure present in *Prevotella copri, Porphyromonas gingivalis,* and *Bacteroides vulgatus.* Branching patterns and acyl chain distribution of lipid structures shown are possible representative structures and are not confirmed.

# Reporting Summary

Nature Research wishes to improve the reproducibility of the work that we publish. This form provides structure for consistency and transparency in reporting. For further information on Nature Research policies, see our Editorial Policies and the Editorial Policy Checklist.

## Statistics

For all statistical analyses, confirm that the following items are present in the figure legend, table legend, main text, or Methods section.

| n/a | Confirmed | |
|---|---|---|
| ☐ | ☒ | The exact sample size (*n*) for each experimental group/condition, given as a discrete number and unit of measurement |
| ☐ | ☒ | A statement on whether measurements were taken from distinct samples or whether the same sample was measured repeatedly |
| ☐ | ☒ | The statistical test(s) used AND whether they are one- or two-sided<br>*Only common tests should be described solely by name; describe more complex techniques in the Methods section.* |
| ☒ | ☐ | A description of all covariates tested |
| ☐ | ☒ | A description of any assumptions or corrections, such as tests of normality and adjustment for multiple comparisons |
| ☐ | ☒ | A full description of the statistical parameters including central tendency (e.g. means) or other basic estimates (e.g. regression coefficient) AND variation (e.g. standard deviation) or associated estimates of uncertainty (e.g. confidence intervals) |
| ☐ | ☒ | For null hypothesis testing, the test statistic (e.g. *F*, *t*, *r*) with confidence intervals, effect sizes, degrees of freedom and *P* value noted<br>*Give P values as exact values whenever suitable.* |
| ☒ | ☐ | For Bayesian analysis, information on the choice of priors and Markov chain Monte Carlo settings |
| ☒ | ☐ | For hierarchical and complex designs, identification of the appropriate level for tests and full reporting of outcomes |
| ☒ | ☐ | Estimates of effect sizes (e.g. Cohen's *d*, Pearson's *r*), indicating how they were calculated |

*Our web collection on statistics for biologists contains articles on many of the points above.*

## Software and code

Policy information about availability of computer code

| | |
|---|---|
| Data collection | For mass spectrometry, Skyline version 21.1 and Compass Data Analysis version 4.3 were used.<br>NCBI Blast+ V2.11.0 was used for homology searches. |
| Data analysis | Mass spectrometry: Spectra were processed with MassLynx (V4.1).<br>RNA-seq: Quality assessment of reads was performed using FastQC (V0.11.8) pre- and post-quality filtering with bbduk (V38.90). Reads were aligned with bowtie2 (V2.3.5.1) and assigned using htseq-count (V0.11.2). Differential expression analysis was performed with EdgeR (V3.32.1) and limma (V3.46.0). Heatmaps were generated with pheatmap (V1.0.12).<br>Crystallography: The data set was integrated with XIA2 (V0.3.7.0) using DIALS (V3.9.1) and scaled with Aimless (V0.3.6). The space group was confirmed with Pointless (V1.10.20). The phase problem was solved by molecular replacement with Phaser (V2.7.17). The model was refined with refmac (V5.8) and manual model building with COOT (V0.1.2). The model was validated using COOT and Molprobity (V4.5). Figures were made with ChimeraX (V1.2). |

For manuscripts utilizing custom algorithms or software that are central to the research but not yet described in published literature, software must be made available to editors and reviewers. We strongly encourage code deposition in a community repository (e.g. GitHub). See the Nature Research guidelines for submitting code & software for further information.

## Data

Policy information about availability of data

All manuscripts must include a data availability statement. This statement should provide the following information, where applicable:

- Accession codes, unique identifiers, or web links for publicly available datasets
- A list of figures that have associated raw data
- A description of any restrictions on data availability

The BT_1526 MIPS structure analyzed during the current study is available in the Protein Data Bank repository, PDB ID:7NWR. RNA-seq reads and data are available at NCBI GEO (https://www.ncbi.nlm.nih.gov/geo/) under accession number GSE193734. Mass spectrometry files, and all unique strains generated in this study are available from the corresponding author upon request. All remaining data generated during this study are included in this published article and its supplementary information files.

# Field-specific reporting

Please select the one below that is the best fit for your research. If you are not sure, read the appropriate sections before making your selection.

☒ Life sciences ☐ Behavioural & social sciences ☐ Ecological, evolutionary & environmental sciences

For a reference copy of the document with all sections, see nature.com/documents/nr-reporting-summary-flat.pdf

# Life sciences study design

All studies must disclose on these points even when the disclosure is negative.

| | |
|---|---|
| Sample size | The mouse experiment used n=8 mice per strain or competition pair tested, to minimize the number of mice used in a pilot test. We initially planned to use n=12 mice per group, in alignment with the mouse group size our lab has previously used for statistical testing of bacterial lipid-dependent physiological effects (1). We obtained however ethical approval for n=8 mice per group, as this was a pilot experiment, and this group size was sufficient to make our final conclusions. Capsule analysis was performed with n=3 capsule extractions per strain to compromise between cost of analysis and low overall abundance of the target of interest. On the basis of past studies showing significant results with the given sample size (2-4), RNA-seq analysis was performed with two biological replicates per strain, which we deemed to be sufficient due to low variability between replicates.<br>1. Johnson, E. L. et al. Sphingolipids produced by gut bacteria enter host metabolic pathways impacting ceramide levels. Nat. Commun. 11, 2471 (2020)<br>2. McNulty, N. P. et al. Effects of diet on resource utilization by a model human gut microbiota containing Bacteroides cellulosilyticus WH2, a symbiont with an extensive glycobiome. PLoS Biol. 11, e1001637 (2013)<br>3. Kijner, S., Cher, A. & Yassour, M. The Infant Gut Commensal Bacteroides dorei Presents a Generalized Transcriptional Response to Various Human Milk Oligosaccharides. Front. Cell. Infect. Microbiol. 12, 854122 (2022)<br>4. Dodd, D., Moon, Y.-H., Swaminathan, K., Mackie, R. I. & Cann, I. K. O. Transcriptomic Analyses of Xylan Degradation by Prevotella bryantii and Insights into Energy Acquisition by Xylanolytic Bacteroidetes. J. Biol. Chem. 285, 30261–30273 (2010) |
| Data exclusions | No data were excluded from the analyses. |
| Replication | With the exception of the mouse colonization, MIPS crystallization, electron microscopy, and RNA-seq, each experiment was performed a minimum of two times. Using the mouse samples, cecal colonization and in mouse-competition analyses were also performed twice. All attempts at replication were successful and supported the conclusions in the manuscript. |
| Randomization | For in vitro experiments, following strain generation, identical treatments were performed (e.g., lipid extraction and analysis, capsule extraction and analysis, AMP resistance and growth curves) so randomized allocation was not necessary, as a physical value was measured which is not influenced by the observer. Growth phase, temperature, and medium for in vitro bacterial growth were controlled and all conditions of a given experiment were performed in parallel. Electron microscopy images chosen are representative of the cell population of that strain. For mouse experiments, mice were randomly assigned to a treatment condition to randomize age and litter of origin. |
| Blinding | The investigators were not blinded during data collection. In the mouse experiment, the same investigators inoculated the mice and collected and analyzed downstream samples, preventing blinding. During computational analysis (RNA-seq), all data were subjected to the same analysis pipelines regardless of condition. During in vitro work with the strains, blinding was not used as the work was performed by a single investigator. |

# Reporting for specific materials, systems and methods

We require information from authors about some types of materials, experimental systems and methods used in many studies. Here, indicate whether each material, system or method listed is relevant to your study. If you are not sure if a list item applies to your research, read the appropriate section before selecting a response.

## Materials & experimental systems

| n/a | Involved in the study |
|-----|----------------------|
| ☒ | ☐ Antibodies |
| ☒ | ☐ Eukaryotic cell lines |
| ☒ | ☐ Palaeontology and archaeology |
| ☐ | ☒ Animals and other organisms |
| ☒ | ☐ Human research participants |
| ☒ | ☐ Clinical data |
| ☒ | ☐ Dual use research of concern |

## Methods

| n/a | Involved in the study |
|-----|----------------------|
| ☒ | ☐ ChIP-seq |
| ☒ | ☐ Flow cytometry |
| ☒ | ☐ MRI-based neuroimaging |

# Animals and other organisms

Policy information about studies involving animals; ARRIVE guidelines recommended for reporting animal research

| | |
|---|---|
| Laboratory animals | Female germ-free C57BL/6 mice aged 4-6 weeks, housed with a 12 hour light:dark cycle at 22±2 deg. C and ~55±10% humidity |
| Wild animals | The study did not involve wild animals. |
| Field-collected samples | The study did not utilize field-collected samples. |
| Ethics oversight | The mouse experiment was performed in accordance with the German legislation on protection of animals with permission to conduct the study obtained from the regional animal welfare committee of the Eberhard Karls Universität Tübingen, registration number EB 03/21 M. |

Note that full information on the approval of the study protocol must also be provided in the manuscript.

