## [Peer Review File · Nature Microbiology]

Peer Review Information

Journal: Nature Microbiology

Manuscript Title: Characterization of inositol lipid metabolism in gut-associated Bacteroidetes

Corresponding author name(s): Ruth Ley

Reviewer Comments & Decisions:

Decision Letter, initial version:

Dear Ruth,

Thank you for your patience while your manuscript "Inositol lipid synthesis is widespread in host-associated Bacteroidetes" was under peer-review at Nature Microbiology. It has now been seen by 3 referees, whose expertise and comments you will find at the end of this email. Although they find your work of some potential interest, they have raised a number of concerns that will need to be addressed before we can consider publication of the work in Nature Microbiology.

In particular, referee #1 asks that you perform pulse-chase labelling experiments to confirm the proposed pathway and both referees #1 and #3 have questions regarding the role of the bacterial capsule. Referee #3 also has an important concern regarding the biological significance of this pathway for bacterial physiology and/or host interactions. We would need additional experimental data to address this point. If you cannot address the questions regarding biological relevance, we would be happy to engage editors at other journals such as Nature Communications to see if a revision without these data would be sufficient.

Should further experimental data allow you to address these criticisms, we would be happy to look at a revised manuscript.

Please include a data availability statement as a separate section after Methods but before references, under the heading "Data Availability". This section should inform readers about the availability of the data used to support the conclusions of your study. This information includes accession codes to public repositories (data banks for protein, DNA or RNA sequences, microarray, proteomics data etc...), references to source data published alongside the paper, unique identifiers such as URLs to data

2repository entries, or data set DOIs, and any other statement about data availability. At a minimum, you should include the following statement: "The data that support the findings of this study are available from the corresponding author upon request", mentioning any restrictions on availability. If DOIs are provided, we also strongly encourage including these in the Reference list (authors, title, publisher (repository name), identifier, year). For more guidance on how to write this section please see:

<http://www.nature.com/authors/policies/data/data-availability-statements-data-citations.pdf>

* If you have not done so already we suggest that you begin to revise your manuscript so that it conforms to our Article format instructions at <http://www.nature.com/nmicrobiol/info/final-submission>. Refer also to any guidelines provided in this letter.

When submitting the revised version of your manuscript, please pay close attention to our [href="https://www.nature.com/nature-research/editorial-policies/image-integrity">Digital Image Integrity Guidelines. and to the following points below:](https://www.nature.com/nature-research/editorial-policies/image-integrity)

{redacted}

Note: This url links to your confidential homepage and associated information about manuscripts you may have submitted or be reviewing for us. If you wish to forward this e-mail

2to co-authors, please delete this link to your homepage first.

Nature Microbiology is committed to improving transparency in authorship. As part of our efforts in this direction, we are now requesting that all authors identified as 'corresponding author' on published papers create and link their Open Researcher and Contributor Identifier (ORCID) with their account on the Manuscript Tracking System (MTS), prior to acceptance. This applies to primary research papers only. ORCID helps the scientific community achieve unambiguous attribution of all scholarly contributions. You can create and link your ORCID from the home page of the MTS by clicking on 'Modify my Springer Nature account'. For more information please visit www.springernature.com/orcid.

If you wish to submit a suitably revised manuscript we would hope to receive it within 6 months. If you cannot send it within this time, please let us know. We will be happy to consider your revision, even if a similar study has been accepted for publication at Nature Microbiology or published elsewhere (up to a maximum of 6 months).

With best wishes,

{redacted}

Reviewer Expertise:

Referee #1: inositol/lipid metabolism
Referee #2: bacterial structural biology
Referee #3: Bacteroidetes, microbiome, lipid metabolism

Reviewer Comments:

Reviewer #1 (Remarks to the Author):

This study provides evidence that the synthesis of phosphatidylinositol (PI) and inositolphosphoceramide (IPC) – previously thought to be restricted to a limited number of bacterial genera - is common in the Bacteroidetes and other members of the gut microbiota. The authors identified a gene cluster in the human gut commensal, *Bacteroides thetaiotaomicron*, that encoded all of the enzymes needed for the atypical PI biosynthetic pathway previously identified in mycobacteria. This cluster was shown to encode a putative MIPS (convert Gl6P to InoP), PIP synthase (convert InoP + CDP-DAG to PIP) and PIP phosphatase (PIP to PI) based on analysis of the levels of intermediates in individual gene knock-out lines. A fourth gene in this cluster, BT_1522, shared homology to yeast IPC synthase and was shown to catalyze a head group exchange reaction to form IPC (PI-DHC), while the function of a fifth gene (BT_1524) remains to be characterized. The authors determined the X-ray structure of BT MIPS, and showed that loss of this enzyme leads to dysregulation in capsule synthesis. The latter phenotype appears to be independent of PI-DHC synthesis and was associated with loss of

3incorporation of inositol into these polysaccharides, suggesting that Ino/InoP or PI is covalently linked to the capsular polysaccharide. The authors also provide indirect evidence for a second pathway of PI synthesis in other PI-synthesizing Bacteroidetes that lack this gene locus. Based on bioinformatics analyses it is proposed that this alternative pathway involves the conversion of inositol-P to CDP-inositol, which then combines with DAG to form PI, while the formation of PI-DHC is catalyzed by a putative haloacid dehalogenase. Genome analysis suggested that three-quarters of the Bacteroides examined had either the PIP or CDP-inositol pathways of PI/IPC synthesis. Overall, this study provides the first definitive characterization of all the steps in the prokaryotic PIP pathway, while also highlighting unanticipated complexity of PI/IPC synthesis in bacteria (outside the mycobacteria) and the potential importance of these phospholipids in capsule formation. This work is likely to be of broad interest to the readership of Nature Microbiology. However, the following comments should be addressed if possible.

Major Comments

The proposed sequence of reactions in the BT PI biosynthetic pathway is largely based on analysis of the steady state levels of key intermediates in different knock-out lines. While the data are consistent with the proposed pathway (and that proposed earlier in mycobacteria), unequivocal evidence for this pathway would require pulse-chase labelling experiments with ^{14}C - or ^{13}C -glucose to show sequential labelling of InoP, PIP, then PI. This is particularly important as PIP is not detected in WT and several of the mutant lines (ΔBT_{1523} , 1526). While it is possible that PIP is rapidly converted to PI in these lines, as proposed, identification of a transiently labelled species would confirm this hypothesis and provide direct evidence for the proposed pathway.

Can the authors discount the possibility that exogenous inositol is directly converted to InoP by an endogenous kinase (as proposed to occur in mycobacteria)?

Can the authors comment on whether inositol, PI or PIP is attached to the capsular polysaccharide and the nature of the linkage? Have they, for example, investigated whether the capsular polysaccharide contains covalently-linked fatty acids after exhaustive solvent extraction?

Fig 2. Have the authors measured the levels of inositol-P in the different mutant lines (ΔBT_{1522} , 1523, 1524, 1525 lines)? Measurement of the levels of this metabolite would allow assessment of whether it functions exclusively as a precursor in the PIP pathway, or whether it is also a functionally important end-product (as occurs in eukaryotes).

Did the authors look at inositol incorporation into capsular polysaccharide in the ΔBT_{1524} knock-out line? If this gene encodes a putative flippase, as proposed, attachment of Inositol-P/PI or PIP to the capsular polysaccharide might be disrupted in this line.

Minor comments

Figure 2. panel A. standard lanes lacks PIP (indicated in legend)

Reviewer #2 (Remarks to the Author):

4Inositol lipids play important roles in eukaryotic cells during cell signaling and membrane homeostasis. The role of inositol lipids in bacteria on the other hand is not well characterized, and the pathways for inositol lipid synthesis are ambiguous. Here, Heaver and colleagues have studied the gene cluster responsible for inositol lipid synthesis in *Bacteroides thetaiotaomicron* (BT). They further solved the structure of one of the proteins in this gene cluster, the myo-inositol-phosphate (MIP) synthase. The authors performed transcriptomic analysis of the inositol pathway in BT and used bioinformatic tools to reveal a novel second putative pathway for bacterial PI synthesis.

The authors present an impressive amount of work characterizing this new pathway in BT, using several different techniques. I have a few suggestions and questions for the authors, see below.

Do the authors have a particular reason to show the genes in figure 1b running from right to left instead of the more conventional way of showing them left to right? I assume the numbers at the top of the figure represent the chromosomal numbering, but this does not seem to be explained in the figure legend and therefore I think requires clarification.

Figure 3: I don't think the figure C(ii) shows very clearly that this is a tetramer. I see only three different colors; it might be helpful to color each subunit in a unique color and show a couple of different views (for example rotated 90 degrees).

I wasn't able to access the tables anywhere, there is a page at the end showing what tables are included, but I couldn't see them.

I find figure 4 somewhat confusing. I see there are replicates for almost all the conditions tested (except ISPT - 0 A and ISPT - 0.2 B, that is not a real replicate). Most of these replicates are placed next to each other, but some are placed far apart, this makes it confusing to orient yourself in the figure, I suggest at least having duplicates next to each other.

I was surprised to see that *F. major* does not produce inositol lipids. Do the authors have an explanation for this observation?

Reviewer #3 (Remarks to the Author):

Overall

In this manuscript, Heaver et al., investigated inositol lipid biosynthesis in gut commensal microbes. They first identified AUR1(PI-Cer synthase) orthologue in *B. theta* (BT1522) and found a gene cluster which was characterized as responsible genes for PI-DAG and PI-DHC. Individual KO in the cluster were generated, confirming individual biosynthesis steps proposed. Authors also found the loss of MIPS(BT1526) causes change in capsule biosynthesis gene cluster expression and may change membrane polysaccharide structure. Finally, PI-DHCs are identified in multiple *Bacteroidales* and more than one gene cluster seems to exist for PI lipid biosynthesis.

5Unfortunately, structure of the manuscript is rather confusing, which makes it hard to understand the message authors want to deliver. Biological relevance (either for the direct benefit of the bacteria or in the context of host-microbiota, considering many of producer species are gut commensal) of phosphatidyl inositol and phosphoinositoyl-DHC is only speculative, not observing any difference between strains deficient or sufficient with PI-DAG or PI-DHC.

Major/scientific points

1. What is the true role of PI lipids in capsule/exopolysaccharide biosynthesis in Bacteroidetes? Results are scattered through the manuscript, such as 1) PI is a component of capsule (which is not clear-see below), 2) PI lipid KO strains have upregulated capsule biosynthesis. These two are not very coherent to each other-1) means PI is a building block and 2) means PI is a regulator. It is not impossible that both are true, but hard to drive the reason why it should be. Is the amount or composition of the capsule changed (along with Fig4, Fig5A) in these KOs? Need more quantitative data.
2. In addition, is there capsule-independent impact to the organism by inositol lipids? Is there any difference in growth, AMP resistance and host colonization, etc? All of these are routinely investigated in commensal species KO.
3. It is interesting to identify two functionally similar gene clusters exist. I am not asking to characterize entire enzymes in the alternative cluster, however, some molecular level explanation is needed. For example, if the alternative cluster does not have BT1525 orthologue, how is the PI linked to DAG? Does the BT1523 homolog not use PIP? If so, do species with alternative cluster not have PIP-conjugated intermediates (phosphatidylinositol-P, as shown in figure 1A)?
4. Throughout the manuscript, all the glycerolipids and sphingolipids are drawn with specific chain length and w-2 (isomethyl) branching. Are any of them confirmed by free fatty acid composition analysis, MS/MS, and/or NMR? I cannot find a MS/MS fingerprint fragment of specific acyl chain length, which can unambiguously assign the chain length distribution. I understand it may not be easy to find the fragment, because loss of phosphoinositol is a dominant fragmentation in these lipids, nonetheless, drawing lipid structure without such data is rather speculative (and might be incorrect).
5. (Fig 2B) Why BT_1526 complementation does not work in *B. thetaiotaomicron*?
6. Line 188: Is this claim supported by any data? Only gene expression changes are shown, not the capsule itself.

Minor/technical points

1. There are several errors in Fig 1A, should authors follow common rule of balanced (bio)chemical reaction (I think authors should, at least for the right half of the figure). The second step of the biosynthesis where BT_1523 is involved, CMP must be written as a released product, as Pi is released at the next step. The fourth step (BT_1522) is also somewhat confusing-Is BT1522 an enzyme using

6phosphatidylinositol as a substrate, exchanging its lipid from DAG to DHC, releasing DAG as a product? If it is the case, release of DAG should be written as a product.

2. Resolution issue for multiple figures - Fig 3ABC, SuppFig3B, SuppFig5, SuppFig6

3. Line 26-27 Brown et al. reported inositol sphingolipids in *B. thetaiotaomicron* and *B. ovatus*.

4. Fig. 2D. What does '100uL(not ug?) capsule' means? If it is a typo of mass unit, please correct. If it really meant to be volume unit, I don't think it is a quantitative result-how much real capsule is in the solution?

5. Fig2D and Fig5A seem to be misplaced-2D is not even mentioned until line 227, after Figure 4 is described. Figure legends do not properly describe these results. These two could form a separate figure (with some additional data on capsule biosynthesis).

6. The nomenclature of lipid is not consistent, hence confusing. For example, in figure 2A, "PE" means phosphatidyl ethanolamine (hence the entire lipid molecule) but in figure 2C, "PE" is phosphoethanolamine group (hence the head group of the lipid) and the entire lipid is described as PE-glycerolipid or PE-DHC, etc. These should be cleared-I would recommend spelling out lipid portion, such as PE-DAG / PE-DHC, clarifying what kind of lipid is conjugated to the head group.

7. From line 186, as well as in the label of Figure 4, annotation of KO strain is also confusing. If a certain strain is generated on the iSPT background, it should be described so (so to speak, iSPTdBT_1626). KO strain is not a WT anymore, so 'WTdBT_1626' is not right-'dBT_1626' is.

8. Line 315: I don't understand the logic that m/z 259 with or without m/z 241 in MS/MS can distinguish structural difference of the fragment. m/z 259 is a signature of PI head group and 241 is a neutral loss of it. Are authors claiming that 259 without 241 is originated from different type of head group? If so, please elaborate.

Author Rebuttal to Initial comments

Reviewer Expertise:

Referee #1: inositol/lipid metabolism

Referee #2: bacterial structural biology

Referee #3: Bacteroidetes, microbiome, lipid metabolism

Reviewer Comments:

7Reviewer #1 (Remarks to the Author):

This study provides evidence that the synthesis of phosphatidylinositol (PI) and inositolphosphoceramide (IPC) – previously thought to be restricted to a limited number of bacterial genera - is common in the Bacteroidetes and other members of the gut microbiota. The authors identified a gene cluster in the human gut commensal, *Bacteroides thetaiotaomicron*, that encoded all of the enzymes needed for the atypical PI biosynthetic pathway previously identified in mycobacteria. This cluster was shown to encode a putative MIPS (convert Gl6P to InoP), PIP synthase (convert InoP + CDP-DAG to PIP) and PIP phosphatase (PIP to PI) based on analysis of the levels of intermediates in individual gene knock-out lines. A fourth gene in this cluster, BT_1522, shared homology to yeast IPC synthase and was shown to catalyze a head group exchange reaction to form IPC (PI-DHC), while the function of a fifth gene (BT_1524) remains to be characterized. The authors determined the X-ray structure of BT

MIPS, and showed that loss of this enzyme leads to dysregulation in capsule synthesis. The latter phenotype appears to be independent of PI-DHC synthesis and was associated with loss of incorporation of inositol into these polysaccharides, suggesting that Ino/InoP or PI is covalently linked to the capsular polysaccharide. The authors also provide indirect evidence for a second pathway of PI synthesis in other PI-synthesizing Bacteroidetes that lack this gene locus. Based on bioinformatics analyses it is proposed that this alternative pathway involves the conversion of inositol-P to CDP-inositol, which then combines with DAG to form PI, while the formation of PI-DHC is catalyzed by a putative haloacid dehalogenase. Genome analysis suggested that three-quarters of the *Bacteroides* examined had either the PIP or CDP-inositol pathways of PI/IPC synthesis. Overall, this study provides the first definitive characterization of all the steps in the prokaryotic PIP pathway, while also highlighting unanticipated complexity of PI/IPC synthesis in bacteria (outside the mycobacteria) and the potential importance of these phospholipids in capsule formation.

This work is likely to be of broad interest to the readership of Nature Microbiology. However, the following comments should be addressed if possible.

Thank you for the interest in our work. Our responses are in red.

Major Comments

The proposed sequence of reactions in the BT PI biosynthetic pathway is largely based on analysis of the steady state levels of key intermediates in different knock-out lines. While the data are consistent with the proposed pathway (and that proposed earlier in mycobacteria), unequivocal evidence for this pathway

8would require pulse-chase labelling experiments with ^{14}C - or ^{13}C -glucose to show sequential labelling of InoP, PIP, then PI. This is particularly important as PIP is not detected in WT and several of the mutant lines (ΔBT_{1523} , 1526). While it is possible that PIP is rapidly converted to PI in these lines, as proposed, identification of a transiently labelled species would confirm this hypothesis and provide direct evidence for the proposed pathway.

We agree that tracking the lipid intermediates dynamically would strengthen the identification of the pathway. We attempted the suggested experiment: we used ^{13}C -glucose to label the inositol headgroup of the lipids in the WT and ΔBT_{1525} strains. Unfortunately, we observed that the ^{13}C -glucose did not quickly incorporate into the inositol headgroup, despite using cultures in late-exponential phase (RR Fig. 1). We increased the length of the incubation to increase uptake of the labeled glucose, but ^{13}C -glucose was instead incorporated into the entire lipid molecule, preventing a clean, identifiable mass shift for each lipid structure (RR Fig. 2-4). Due to the lack of uptake of exogenous inositol (see below), this also precluded providing labeled inositol as a trackable substrate.

9spectra of lipid species in WT or ΔBT_{1525} strains grown in minimal medium with standard glucose, then washed and passaged into minimal medium with standard glucose or D-glucose-13C6 (Sigma #389374) for 13 minutes. Cells were immediately moved into lipid extraction solvent following 13 minute incubation at 37 deg C.

Had we been able to track inositol using this isotope labeling technique, we still would not be able to observe the transition between all inositol lipids of interest (PIP-DAG, PI-DAG, PI-DHC). There is no strain in which all inositol lipid metabolites are simultaneously present to allow tracking through each successive lipid species. WT BT has detectable PI-DAG and PI-DHC, but no PIP-DAG. Δ BT_1525 has detectable PIP-DAG, but no PI-DAG or PI-DHC.

Though we were unable to show the inositol lipid transformations dynamically, a PIP-DAG intermediate is the most likely explanation for the exclusive presence of PIP-DAG in the Δ BT_1525 knockout strain and follows the expectation from existing literature. BT_1525 has homology to the well established PIP-DAG synthase in *Mycobacterium tuberculosis* ("PgsA1," e-value 3-22, 30.0% identity). Logically, if PI-DAG was phosphorylated to produce PIP-DAG, and PI-DAG used to synthesize PI-DHC, there is no single knockout condition that would cause the accumulation of PIP-DAG without the simultaneous presence of PI-DHC. As such, even without the dynamic tracking, the data support the pathway proposed.

12Can the authors discount the possibility that exogenous inositol is directly converted to InoP by an endogenous kinase (as proposed to occur in mycobacteria)?

This is a very good question. To address this question, we grew wild-type and Δ BT_1526 BT in minimal medium with solely glucose as the carbon source, or a 1:1 mix of glucose:inositol. A detectable, but very minute amount of PI was synthesized in the Δ BT_1526 strain grown in the inositol mix, but not when grown in glucose (main text Fig. 3D; page 9). However, the intensity of this PI signal (of Δ BT_1526 grown with inositol) was \sim 10x lower than the WT strain grown in either medium, despite the large molar abundance of inositol in the medium. The uptake of exogenous inositol appears to happen at a very, very low level, but appears insufficient for wild-type levels of inositol lipid synthesis in the Δ BT_1526 strain. Our results indicate that BT is very unlikely to assimilate inositol from its environment.

Can the authors comment on whether inositol, PI or PIP is attached to the capsular polysaccharide and the nature of the linkage? Have they, for example, investigated whether the capsular polysaccharide contains covalently-linked fatty acids after exhaustive solvent extraction?

This insightful question gets at whether PI or PIP act as a physical anchor of the capsule.

One way to answer this question is to expand our lipid analysis to include more polar lipid species that are extracted in the aqueous phase of a Folch lipid extraction, the phase in which a heavily glycosylated lipid would be found. The loss of a glycosylated lipid in an inositol lipid knockout could indicate the presence of an inositol lipid-bound polysaccharide, suggesting a direct role for these lipids in anchoring capsular components to the membrane. We extracted aqueous-phase glycolipids and separated these lipids by TLC, detected lipids with the reagent primuline, and subsequently sprayed the plate with orcinol to detect glycans. All strains (WT, Δ BT_1522, Δ BT_1523, Δ BT_1524, Δ BT_1525, Δ BT_1526, and Δ BT_0870, which lacks sphingolipids) produced glycosylated lipids, indicating a potential capsular component, which was likely lipooligosaccharide. However, Δ BT_1522 and Δ BT_1525 appear to produce an additional glycosylated lipid, suggesting that PIP may be a substrate for glycosylation (RR. Fig. 5). We find that determination of this carbohydrate structure is beyond the scope of this paper but represents an exciting foundation for follow-up research.

higher magnification. Arrows indicate band enriched in ΔBT_{1522} and ΔBT_{1525} .

Echoing these findings, In our analyses of capsule components, we extracted the capsule with 1% phenol, then performed a clean-up step with DNase, RNase, and protease, followed by dialysis. Fatty acids remained detectable in these samples and were present regardless of each strain's production of inositol lipids (Supp. Fig. 8E; page 56). This observation suggests that these fatty acids are independent of inositol lipids, and may be derived from lipooligosaccharide co-extracted in the samples.

In revision, we vastly expanded our capsule analysis to include every knockout strain in the inositol lipid cluster and all glycosyl residues we could measure (11, including inositol) (see response to Reviewer 3 below). Our capsular monosaccharide analysis showed inositol is a very minor component of the capsule, at maximum 0.2% molar abundance of glycosyl residues detected (main text Fig. 4B; page 12). While it is possible that this small amount of inositol could be lipid-derived and covalently bound to a capsular glycan structure, the evidence is insufficient to implicate the fitness defect of the inositol lipid-deficient strain *in vitro* and *in vivo* as due the presence of inositol as a structural component of the capsule.

In addition to any putative inositol glycolipids acting as capsule components, inositol was detected in the ΔBT_{1523} strain lacking inositol lipids, indicating that not all inositol in the capsule is lipid-bound (page 12).

Fig 2. Have the authors measured the levels of inositol-P in the different mutant lines (ΔBT_{1522} , 1523, 1524, 1525 lines)? Measurement of the levels of this metabolite would allow assessment of whether it functions exclusively as a precursor in the PIP pathway, or whether it is also a functionally important end-product (as occurs in eukaryotes).

This is an interesting suggestion. In response, we have extracted total metabolites from WT, ΔBT_{1522} , ΔBT_{1523} , ΔBT_{1524} , ΔBT_{1525} , and ΔBT_{1526} strains in duplicate to quantify inositol phosphate levels by LC-MS/MS. Washed bacteria of each strain (in duplicate) were sonicated into a methanol/acetonitrile/formic acid mix and analyzed using a slightly modified protocol from Martano *et al.*¹. We acquired MS/MS data in multiple reaction monitoring (MRM) mode using a fragmentation energy of 40 eV, to distinguish inositol-phosphate from G6P.

We were able to detect standards of inositol phosphate and its mass partner, glucose-6-phosphate, with a limit of detection of 0.04 ug/mL for inositol phosphate (RR Fig 6). However, no free inositol phosphate or

15glucose-6-phosphate was detected in any sample. We expect this is due to the fast downstream metabolism of these metabolites into more complex structures.

Did the authors look at inositol incorporation into capsular polysaccharide in the Δ BT_1524 knock-out line? If this gene encodes a putative flippase, as proposed, attachment of Inositol-P/PI or PIP to the capsular polysaccharide might be disrupted in this line.

Thank you for this suggestion. We have repeated the capsular polysaccharide analysis with $n=3$ for each knockout strain in the inositol lipid cluster. Inositol was in fact detectable in the capsule of the Δ BT_1524 strain (main text Fig. 4B; page 12), roughly at equivalent levels to the WT strain, and there was no other notable difference in Δ BT_1524 capsular glucosyl residues from WT (Supp. Fig. 8; page 56). The Δ BT_1524 strain also had a similar growth trend (Supp. Fig. 9; page 57) and susceptibility to the antimicrobial peptide LL-37 (main text Fig. 4C; page 12) as WT BT; the function of BT_1524 remains unclear.

Minor comments

16Figure 2. panel A. standard lanes lacks PIP (indicated in legend)

We thank the reviewer for bringing this oversight to our attention. In the solvent system in this panel, PIP does not migrate above baseline. We have removed this standard from the legend.

Reviewer #2 (Remarks to the Author):

Inositol lipids play important roles in eukaryotic cells during cell signaling and membrane homeostasis. The role of inositol lipids in bacteria on the other hand is not well characterized, and the pathways for inositol lipid synthesis are ambiguous. Here, Heaver and colleagues have studied the gene cluster responsible for inositol lipid synthesis in *Bacteroides thetaiotaomicron* (BT). They further solved the structure of one of the proteins in this gene cluster, the myo-inositol-phosphate (MIP) synthase. The authors performed transcriptomic analysis of the inositol pathway in BT and used bioinformatic tools to reveal a novel second putative pathway for bacterial PI synthesis.

The authors present an impressive amount of work characterizing this new pathway in BT, using several different techniques. I have a few suggestions and questions for the authors, see below.

Thank you for these supportive comments.

Do the authors have a particular reason to show the genes in figure 1b running from right to left instead of the more conventional way of showing them left to right? I assume the numbers at the top of the figure represent the chromosomal numbering, but this does not seem to be explained in the figure legend and therefore I think requires clarification.

Thank you for pointing this out. We have reversed the direction to show the genes running from left to right, and clarified in the figure legend that these numbers indicate chromosomal numbering. This is available on page 3.

Figure 3: I don't think the figure C(ii) shows very clearly that this is a tetramer. I see only three different colors; it might be helpful to color each subunit in a unique color and show a couple of different views (for example rotated 90 degrees).

17We have taken this suggestion and recolored the figure, increasing the number of views from different angles and adding cartoon representations to clarify the structure. This is available on page 9.

I wasn't able to access the tables anywhere, there is a page at the end showing what tables are included, but I couldn't see them.

We apologize for the difficulty accessing the table and hope they are accessible in this revised version.

I find figure 4 somewhat confusing. I see there are replicates for almost all the conditions tested (except ISPT – 0 A and ISPT – 0.2 B, that is not a real replicate). Most of these replicates are placed next to each other, but some are placed far apart, this makes it confusing to orient yourself in the figure, I suggest at least having duplicates next to each other.

Indeed the order of the samples is determined by similarity in gene expression. The sample columns in this figure are sorted by Euclidean clustering, such that samples with similar expression trends in the dataset are adjacent to one another. In this sense, the location of most biological replicates next to one another is not a pre-determined choice of order, but is rather indicated by the similarity of the expression trends in these samples. We chose to present the data with this clustering method to show the strong clustering of the Δ BT_1526 strain expression compared to the other strains.

I was surprised to see that *F. major* does not produce inositol lipids. Do the authors have an explanation for this observation?

We were also surprised, but there are a few possible explanations. *F. major* is environmentally derived (from an alga) and not a close relative of BT although they are in the same phylum. Though it has genes in its genome with considerable homology to genes in the *B. theta*-like inositol cluster, they have relatively low overall sequence homology (for example, the BT_1525 amino acid homology e-value for *F. major* is 3.0E-28, compared to an average e-value of approximately 3.7E-145 among *Bacteroides* spp. with a BT_1525 homolog). The low homology may reflect a loss or altered function of this cluster in *F. major*. Alternatively, in the Actinobacteria, which make inositol glycerophospholipids (not sphingolipids), the inositol headgroup of the lipid serves as an important polysaccharide anchor. If *F. major* likewise produces a strongly glycosylated lipid from a base inositol lipid, this would not be detected in the organic

phase of the standard Folch lipid extraction performed. Finally, the gene cluster may simply not be active under tested conditions.

Reviewer #3 (Remarks to the Author):

Overall

In this manuscript, Heaven et al., investigated inositol lipid biosynthesis in gut commensal microbes. They first identified AUR1 (PI-Cer synthase) orthologue in *B. theta* (BT1522) and found a gene cluster which was characterized as responsible genes for PI-DAG and PI-DHC. Individual KO in the cluster were generated, confirming individual biosynthesis steps proposed. Authors also found the loss of MIPS (BT1526) causes change in capsule biosynthesis gene cluster expression and may change membrane polysaccharide structure. Finally, PI-DHCs are identified in multiple Bacteroidales and more than one gene cluster seems to exist for PI lipid biosynthesis.

Unfortunately, structure of the manuscript is rather confusing, which makes it hard to understand the message authors want to deliver.

We agree that the structure of the paper could be improved, and our revised version is, we believe, considerably clearer. The results section is now divided into sections with subheadings.

Biological relevance (either for the direct benefit of the bacteria or in the context of host-microbiota, considering many of producer species are gut commensal) of phosphatidyl inositol and phosphoinositoyl-DHC is only speculative, not observing any difference between strains deficient or sufficient with PI-DAG or PI-DHC.

Thank you for pointing out this deficiency. We have addressed this issue with comparative growth assays *in vitro*, with an antimicrobial peptide resistance assay, and with germfree mouse colonization experiments to assess *in-vivo* fitness of the strains. We are excited to report the results of these experiments in the paper, as they show that inositol and inositol lipids are important for host-microbe interactions.

19Major/scientific points

1. What is the true role of PI lipids in capsule/exopolysaccharide biosynthesis in Bacteroidetes? Results are scattered through the manuscript, such as 1) PI is a component of capsule (which is not clear-see below), 2) PI lipid KO strains have upregulated capsule biosynthesis. These two are not very coherent to each other-1) means PI is a building block and 2) means PI is a regulator. It is not impossible that both are true, but hard to drive the reason why it should be. Is the amount or composition of the capsule changed (along with Fig4, Fig5A) in these KOs? Need more quantitative data.

Thank you for pointing out this issue. In our revised version of the manuscript, we have streamlined the section related to capsule gene expression and supplemented these observations with new data where we measure inositol in the capsules of all knockout strains. Together these data offer support for a role of inositol in the structure of the capsule. Given that we have less data related to the role of inositol as a regulator, we have removed this concept from the paper. We find that the new version is clearer and presents a more straightforward interpretation of the data.

To address the question of whether inositol is a structural component of the capsule, we have generated more quantitative data as requested. Specifically, we have repeated the analysis of the glucosyl composition of the capsule with $n=3$ for each knockout strain in the inositol lipid cluster. As a molar proportion of the total glucosyl moieties in the capsule, inositol is a very minor component, comprising at most only 0.2% of the glucosyl residues (and on average, well below this at approx. 0.04%; main text Fig. 4B; page 12). This does not exclude that, like in the Mycobacteria, the inositol may serve as an anchor molecule off of which more extensive carbohydrate structures are bound. As in the response to Reviewer 1 above, we additionally compared glycosylated lipids from each knockout strain by TLC (RR Fig. 5). The major glycosylated lipid band was consistent between each strain, however ΔBT_{1522} and ΔBT_{1525} , which share PIP-DAG synthesis, produced a notable additional glycosylated lipid band that is faintly visible in the WT and ΔBT_{1524} strains, but absent in ΔBT_{1526} . This could suggest that PIP-DAG, which is more abundant in ΔBT_{1522} and ΔBT_{1525} , can be used as a substrate for subsequent glycan modifications, producing more of a glycolipid that is less abundant in WT inositol lipid strains. Pursuing the characterization of these putative inositol glycolipids is beyond the scope of the present paper.

It is also interesting to note that inositol was also detected in the capsule of the ΔBT_{1523} strain, which lacks inositol lipids. This suggests that inositol can in fact be incorporated into the capsule independent of its presence on an inositol lipid, making it a very minor structural building block within the capsule.

As inositol is only a minor capsule component, we can only hypothesize why the loss of inositol changed expression of CPS loci in the RNA-seq experiment. It could be a stochastic effect between bacterial batches, independent of inositol entirely; in the capsule analysis, the molar ratio of each glycosyl residue was in fact very similar between WT and Δ BT_1526 strains. This could result from the activation of multiple CPS loci, creating glycosyl residue profiles that “average out” the impact of an individual CPS locus. Overall, we have reduced our interpretation of the importance of the capsule in the text, and removed text suggesting inositol to function as a regulator in capsule expression.

Alternatively, the loss of inositol lipids would change the overall membrane lipid composition, leading to a more positive membrane charge (see AMP resistance results below). While this resultant more positive membrane charge is likely responsible for the increased resistance to the specific cationic AMP tested, capsular carbohydrates often contribute a negative charge that is protective against phagocytosis, and CPS expression might be regulated to restore this charge.

2. In addition, is there capsule-independent impact to the organism by inositol lipids? Is there any difference in growth, AMP resistance and host colonization, etc? All of these are routinely investigated in commensal species KO.

To address this question, we’ve performed growth curves of all strains in complex and minimal media (Supp. Fig. 9; page 57). Despite normal growth in rich medium, the Δ BT_1526 strain failed to reach the maximum OD of the other strains in minimal medium, suggesting that lack of inositol, but not inositol lipids, reduced fitness when glucose is provided as a sole carbon source.

Additionally, we tested the resistance of each strain against LL-37, a cationic AMP produced in the human gastrointestinal tract (main text Fig. 4C; page 12). As inositol lipids are negatively charged, they (along with phosphatidylserine) contribute a negative net charge to the bacterial membrane. Consistent with the loss of inositol lipids and the resulting increase in membrane charge, the Δ BT_1523 and Δ BT_1526 strains had increased resistance to LL-37. This is noted in the text on page 15.

Finally, to determine the physiological significance of inositol lipids in a mammalian host, we mono-associated WT, Δ BT_1522, Δ BT_1523, and Δ BT_1526 strains in germ-free mice for 14 days. No major difference in cecal colonization was seen for the strains when individually associated (main text Fig. 4D; page 12); however, a competition experiment between WT and Δ BT_1523 strains identified a fitness defect for Δ BT_1523, despite initial inoculation at 3X the concentration of the WT strain (main text Fig. 4E; page 12). This is noted in the text on page 16.

3. It is interesting to identify two functionally similar gene clusters exist. I am not asking to characterize entire enzymes in the alternative cluster, however, some molecular level explanation is needed. For example, if the alternative cluster does not have BT1525 orthologue, how is the PI linked to DAG? Does the BT1523 homolog not use PIP? If so, do species with alternative cluster not have PIP-conjugated intermediates (phosphatidylinositol-P, as shown in figure 1A)?

In the proposed alternative cluster pathway, PIP-DAG would not be an intermediate structure. After submission of our original manuscript, we found another paper identifying CDP-inositol as an intermediate in bacterial PI-dialkylether glycerol synthesis (Jorge et al. 2015, *Env. Micro.*). In this instance in *Rhodothermus marinus*, CDP-inositol is added as a headgroup to dialkylether glycerol, producing a lipid with a PI, not PIP, headgroup. We expect a similar transformation to occur in our proposed alternative inositol lipid pathway, with the predicted CDP-alcohol-phosphatidyltransferase (e.g., BVU_RS13105 in *B. vulgatus*) linking CDP-inositol to DAG and producing PI-DAG, as opposed to PIP-DAG synthesis by BT_1523. There would be one fewer enzyme in the alternate pathway, as there would be no PIP-DAG to be dephosphorylated to PI-DAG. This is noted in the text on page 20. This additional citation is now included in the paper, and we also included a structural pathway comparison in the supplement, comparing the BT-like inositol lipid pathway to our expectation of the alternative inositol lipid cluster (Supp. Fig. 10; page 58).

In wild-type BT, it is almost impossible to detect any PIP-DAG in any strain except the BT_1525 knockout as it is in such low abundance (as mentioned in the manuscript, we expect BT_1525 very quickly catalyzes this conversion). As such, PIP might not be detected even in species that do include it in their pathway. Nonetheless, we performed a PIP-specific lipid extraction on 5 species with homology to the proposed alternative cluster (*Prevotella micans*, *Prevotella nigrescens*, *Prevotella veroralis*, *Phoecicola dorei*, and *Bacteroides vulgatus*), comparing these to WT BT and Δ BT_1525 strains. PIP-DAG was only detectable in the Δ BT_1525 strain, with a limit of detection of 0.85 μ g/mL (RR Figs 7-8). Detected PIP-DAG species included PIP-DAG 29:0, PIP-DAG 30:0, and PIP-DAG 31:0, with estimated concentrations in the extracts of: 5.86, 9.98 and 5.88 μ g/mL, respectively. The estimated concentration normalized to biomass were 2.33, 3.97, and 2.34 μ g/g respectively.

RR Fig 7. MS2 spectra of the PIP-DAG synthetic standard and PIP-DAG 29:0, 30:0, and 31:0 detected in the Δ BT_1525 strain. The diagnostic ions for the PIP headgroup (m/z 241.01 and 320.98) are indicated in red boxes and the structure of PIP-DAG 30:0 (lower right panel). No PIP-DAG could be detected above background in the WT BT strain (lower left panel).

(241.01 and 320.98) were not observed.

4. Throughout the manuscript, all the glycerolipids and sphingolipids are drawn with specific chain length and w-2 (isomethyl) branching. Are any of them confirmed by free fatty acid composition analysis, MS/MS, and/or NMR? I cannot find a MS/MS fingerprint fragment of specific acyl chain length, which can unambiguously assign the chain length distribution. I understand it may not be easy to find the fragment, because loss of phosphoinositol is a dominant fragmentation in these lipids, nonetheless, drawing lipid structure without such data is rather speculative (and might be incorrect).

To better clarify the structures of these lipids, we have performed FAME on total bacterial lipids with and without sphingolipids and on an isolated lipid fraction comprising PI-DAG and PS-DAG. We combined this with MALDI MS mass determinations of lipid structures in a PI-DHC and PI-DAG/PS-DAG fraction, and LC-MS/MS data on the PIP-DAG structures in the Δ BT_1525 strain. In total lipids, as well as the PI-DAG/PS-DAG fraction, *anteiso*-C15 was more abundant than its *iso*- isomer, which is consistent with the original literature describing lipid branching in BT. We have re-drawn the diagrams to reflect these data with the most likely structures. These figures and structural explanations are available in Supp. Fig. 2+3 (pages 43, 48) and Supp. Table 8. For the DHC-based lipids, which we could not conclusively structurally define, we have additionally added this uncertainty to the figure captions where applicable.

5. (Fig 2B) Why BT_1526 complementation does not work in *B. thetaiotaomicron*?

Despite additional experiments, we have no explanation for the failure of the BT_1526 complementation. We have tried it multiple times, on knockouts both in WT and iSPT backgrounds. We have added an additional region upstream of the native BT_1526. The insertion location is identical to that for the remainder of the complements, which were all successful. We have fully sequenced the entirety of the inositol lipid cluster in these BT_1526 complementation strains and cannot find any off-site mutations that could otherwise inhibit inositol lipid synthesis. We have added a short explanation in the text (page 26).

6. Line 188: Is this claim supported by any data? Only gene expression changes are shown, not the capsule itself.

In addition to the more detailed capsule analysis described above, we have changed the wording of this sentence to clarify that any addition of inositol to the capsule is hypothetical (page 11).

25Minor/technical points

1. There are several errors in Fig 1A, should authors follow common rule of balanced (bio)chemical reaction (I think authors should, at least for the right half of the figure). The second step of the biosynthesis where BT_1523 is involved, CMP must be written as a released product, as Pi is released at the next step. The fourth step (BT_1522) is also somewhat confusing-Is BT1522 an enzyme using phosphatidylinositol as a substrate, exchanging its lipid from DAG to DHC, releasing DAG as a product? If it is the case, release of DAG should be written as a product.

Thank you, we have edited the diagram to include substrates and products comprehensively.

2. Resolution issue for multiple figures - Fig 3ABC, SuppFig3B, SuppFig5, SuppFig6

We have now provided higher resolution images for these figures.

3. Line 26-27 Brown et al. reported inositol sphingolipids in *B. thetaiotaomicron* and *B. ovatus*.

We have added *B. ovatus* to *B. theta* in the text.

4. Fig. 2D. What does '100uL(not ug?) capsule' means? If it is a typo of mass unit, please correct. If it really meant to be volume unit, I don't think it is a quantitative result-how much real capsule is in the solution?

Our updated capsule data is presented quantitatively as molar percentage in the capsular extract (pages 12, 56).

5. Fig2D and Fig5A seem to be misplaced-2D is not even mentioned until line 227, after Figure 4 is described. Figure legends do not properly describe these results. These two could form a separate figure (with some additional data on capsule biosynthesis).

26We have edited the figure order to be more thematic and better follow the order of the text (expanding the capsule analysis data).

6. The nomenclature of lipid is not consistent, hence confusing. For example, in figure 2A, “PE” means phosphatidyl ethanolamine (hence the entire lipid molecule) but in figure 2C, “PE” is phosphoethanolamine group (hence the head group of the lipid) and the entire lipid is described as PE-glycerolipid or PE-DHC, etc. These should be cleared- I would recommend spelling out lipid portion, such as PE-DAG / PE-DHC, clarifying what kind of lipid is conjugated to the head group.

Thank you, we have taken this suggestion and renamed the lipids throughout the text and figures. We have additionally changed the lipid nomenclature as it refers to the number of carbons to reflect standard LipidMaps nomenclature (PI-DAG 30:0, instead of C33 PI-DAG).

7. From line 186, as well as in the label of Figure 4, annotation of KO strain is also confusing. If a certain strain is generated on the iSPT background, it should be described so (so to speak, iSPTdBT_1626). KO strain is not a WT anymore, so ‘WTdBT_1626’ is not right-‘dBT_1626’ is.

We have changed the labeling of each strain throughout the text and figures to precisely indicate the background strain in this way.

8. Line 315: I don’t understand the logic that m/z 259 with or without m/z 241 in MS/MS can distinguish structural difference of the fragment. m/z 259 is a signature of PI head group and 241 is a neutral loss of it. Are authors claiming that 259 without 241 is originated from different type of head group? If so, please elaborate.

It could, for instance, be a phosphorylated mannose, glucose, or fructose. *Sphingomonas paucimobilis* is known to make interesting glycosylated sphingolipids, including GSL-1, which has a GlcA headgroup, and GSL-4A, in which the GSL-1 base structure is further modified with GlcN, Gal, and Man monomers. Both of these structures, however, have a GlcA directly attached to the lipid, which does not correspond with a mass of 259. So this fragment could result from the synthesis of some other phospho-hexose SL.

Decision Letter, first revision:

27Dear Ruth,

Thank you for submitting your revised manuscript "Inositol lipid synthesis is widespread in host-associated Bacteroidetes" (NMICROBIOL-21041057A). It has now been seen by the original referees and their comments are below. The reviewers find that the paper has improved in revision, and therefore we'll be happy in principle to publish it in Nature Microbiology, pending minor revisions to satisfy the referees' final requests and to comply with our editorial and formatting guidelines.

Thank you again for your interest in Nature Microbiology Please do not hesitate to contact me if you have any questions.

Sincerely,

{redacted}

Reviewer #1 (Remarks to the Author):

I commend the authors for addressing, or attempting to address, the key points raised by all three reviewers, in particular undertaking the ¹³C-glucose labeling experiments (unsuccessful), measuring inositol uptake, and further assessing the role of PI/PIPs as anchors for the capsular polysaccharide and the potential requirement for this pathway in bacterial colonization of the gut. These changes, together with others recommended in the reviews, significantly strengthen the conclusions and impact of the study. I believe that the findings will be of broad interest to the molecular microbiology community.

Reviewer #2 (Remarks to the Author):

The authors have answered my previous comments satisfactory and made the suggested modifications.

Reviewer #3 (Remarks to the Author):

I appreciate authors' dedicated efforts to improve the manuscript, both in science and in the structure. Most of my questions are addressed properly-only a few remain.

28Figure 4: As authors mentioned, quantitative analysis confirms that inositol is not a part of capsule (component of the repeating unit). As suggested, it still could be a part of the lipid membrane linker/anchor, however, abundance itself is low and statistical power is not enough to prove or disprove it. Also as described, CPS1/6 upregulation is associated with BT_1526, but not with other downstream PI-lipid synthesis genes, which confirms that CPS regulation is only related with inositol phosphate, but not with PI-lipids. It could be a meaningful negative confirmation (probably the title of the figure needs to be modified).

Related to the question, as described in Supp figure 7, BT_1526 KO show dense exopolysaccharides production compared to WT. Is the similar phenotype observed in other downstream KOs, and/or can the amount of the produced CPS in different KOs be compared quantitatively? This might be an alternative explanation, if change of CPS production is observed in other KOs.

Supp figure 12: I understand authors' explanation of the MS/MS patterns, however don't 100% concur that the presence or absence of m/z 241 (non-specific neutral loss ion) can be the definite evidence, to prove phosphohexosyl group is different between two species. I agree it is an attractive (and likely to be a correct) hypothesis that different stereochemistry of the phosphoryl hexose could make different 259/241 abundance ratio-is there any report, showing that different phosphohexosyl head groups could be distinguished by MS/MS, with the neutral loss pattern? If so, that would be great to support the argument (As a reference point, there are a few studies on phosphatidylglucoside (PtdGlc), but not much MS/MS info is reported).

In parallel, regarding the structural differences between *N. acidiphilum* and *S. paucimobilis* hexosylceramides, supp 12A is described as "PI-DHC" and 12B as "a phosphorylated hexose DHC" (but not PI-DHC, based on MS/MS) in the text. If so, structures of 12B should be described and drawn as such (For example, not "PI-DHC 36:0" but "Hex-P-DHC 36:0").

Decision Letter, final checks:

Dear Ruth,

Thank you for your patience as we've prepared the guidelines for final submission of your Nature Microbiology manuscript, "Inositol lipid synthesis is widespread in host-associated Bacteroidetes" (NMICROBIOL-21041057A). Please carefully follow the step-by-step instructions provided in the attached file, and add a response in each row of the table to indicate the changes that you have made. Please also check and comment on any additional marked-up edits we have proposed within the text. Ensuring that each point is addressed will help to ensure that your revised manuscript can be swiftly handed over to our production team.

When you upload your final materials, please include a point-by-point response to any remaining

29reviewer comments.

In recognition of the time and expertise our reviewers provide to Nature Microbiology's editorial process, we would like to formally acknowledge their contribution to the external peer review of your manuscript entitled "Inositol lipid synthesis is widespread in host-associated Bacteroidetes". For those reviewers who give their assent, we will be publishing their names alongside the published article.

Nature Microbiology offers a Transparent Peer Review option for new original research manuscripts submitted after December 1st, 2019. As part of this initiative, we encourage our authors to support increased transparency into the peer review process by agreeing to have the reviewer comments, author rebuttal letters, and editorial decision letters published as a Supplementary item. When you submit your final files please clearly state in your cover letter whether or not you would like to participate in this initiative. Please note that failure to state your preference will result in delays in accepting your manuscript for publication.

Cover suggestions

As you prepare your final files we encourage you to consider whether you have any images or illustrations that may be appropriate for use on the cover of Nature Microbiology.

Nature Microbiology has now transitioned to a unified Rights Collection system which will allow our Author Services team to quickly and easily collect the rights and permissions required to publish your work. Approximately 10 days after your paper is formally accepted, you will receive an email in providing you with a link to complete the grant of rights. If your paper is eligible for Open Access, our Author Services team will also be in touch regarding any additional information that may be required to arrange payment for your article.

30Please note that you will not receive your proofs until the publishing agreement has been received through our system.

Please note that *Nature Microbiology* is a Transformative Journal (TJ). Authors may publish their research with us through the traditional subscription access route or make their paper immediately open access through payment of an article-processing charge (APC). Authors will not be required to make a final decision about access to their article until it has been accepted. [Find out more about Transformative Journals](https://www.springernature.com/gp/open-research/transformative-journals)

Authors may need to take specific actions to achieve [compliance with funder and institutional open access mandates](https://www.springernature.com/gp/open-research/funding/policy-compliance-faqs). If your research is supported by a funder that requires immediate open access (e.g. according to [Plan S principles](https://www.springernature.com/gp/open-research/plan-s-compliance)) then you should select the gold OA route, and we will direct you to the compliant route where possible. For authors selecting the subscription publication route, the journal's standard licensing terms will need to be accepted, including [self-archiving policies](https://www.springernature.com/gp/open-research/policies/journal-policies). Those licensing terms will supersede any other terms that the author or any third party may assert apply to any version of the manuscript.

Please use the following link for uploading these materials:
{redacted}

Best regards,

{redacted}

Reviewer #1:

Remarks to the Author:

I commend the authors for addressing, or attempting to address, the key points raised by all three reviewers, in particular undertaking the 13C-glucose labeling experiments (unsuccessful), measuring inositol uptake, and further assessing the role of PI/PIPs as anchors for the capsular polysaccharide

31and the potential requirement for this pathway in bacterial colonization of the gut. These changes, together with others recommended in the reviews, significantly strengthen the conclusions and impact of the study. I believe that the findings will be of broad interest to the molecular microbiology community.

Reviewer #2:

Remarks to the Author:

The authors have answered my previous comments satisfactory and made the suggested modifications.

Reviewer #3:

Remarks to the Author:

I appreciate authors' dedicated efforts to improve the manuscript, both in science and in the structure. Most of my questions are addressed properly-only a few remain.

Figure 4: As authors mentioned, quantitative analysis confirms that inositol is not a part of capsule (component of the repeating unit). As suggested, it still could be a part of the lipid membrane linker/anchor, however, abundance itself is low and statistical power is not enough to prove or disprove it. Also as described, CPS1/6 upregulation is associated with BT_1526, but not with other downstream PI-lipid synthesis genes, which confirms that CPS regulation is only related with inositol phosphate, but not with PI-lipids. It could be a meaningful negative confirmation (probably the title of the figure needs to be modified).

Related to the question, as described in Supp figure 7, BT_1526 KO show dense exopolysaccharides production compared to WT. Is the similar phenotype observed in other downstream KOs, and/or can the amount of the produced CPS in different KOs be compared quantitatively? This might be an alternative explanation, if change of CPS production is observed in other KOs.

Supp figure 12: I understand authors' explanation of the MS/MS patterns, however don't 100% concur that the presence or absence of m/z 241 (non-specific neutral loss ion) can be the definite evidence, to prove phosphohexosyl group is different between two species. I agree it is an attractive (and likely to be a correct) hypothesis that different stereochemistry of the phosphoryl hexose could make different 259/241 abundance ratio-is there any report, showing that different phosphohexosyl head groups could be distinguished by MS/MS, with the neutral loss pattern? If so, that would be great to support the argument (As a reference point, there are a few studies on phosphatidylglucoside (PtdGlc), but not much MS/MS info is reported).

In parallel, regarding the structural differences between *N. acidiphilum* and *S. paucimobilis* hexosylceramides, supp 12A is described as "PI-DHC" and 12B as "a phosphorylated hexose DHC" (but not PI-DHC, based on MS/MS) in the text. If so, structures of 12B should be described and drawn as such (For example, not "PI-DHC 36:0" but "Hex-P-DHC 36:0").

32Final Decision Letter:

Dear Ruth,

I am pleased to accept your Article "Characterization of inositol lipid metabolism in gut-associated Bacteroidetes" for publication in Nature Microbiology. Thank you for having chosen to submit your work to us and many congratulations.

Acceptance of your manuscript is conditional on all authors' agreement with our publication policies (see <https://www.nature.com/nmicrobiol/editorial-policies>). In particular your manuscript must not be published elsewhere and there must be no announcement of the work to any media outlet until the publication date (the day on which it is uploaded onto our website).

Please note that *Nature Microbiology* is a Transformative Journal (TJ). Authors may publish their research with us through the traditional subscription access route or make their paper immediately open access through payment of an article-processing charge (APC). Authors will not be required to make a final decision about access to their article until it has been accepted. [Find out more about Transformative Journals](https://www.springernature.com/gp/open-research/transformative-journals)

Authors may need to take specific actions to achieve [compliance with funder and institutional open access mandates](https://www.springernature.com/gp/open-research/funding/policy-compliance-faqs). If your research is supported by a funder that requires immediate open access (e.g. according to <https://www.springernature.com/gp/open-research/funding/policy-compliance-faqs>)

33[Plan S principles](https://www.springernature.com/gp/open-research/plan-s-compliance)) then you should select the gold OA route, and we will direct you to the compliant route where possible. For authors selecting the subscription publication route, the journal's standard licensing terms will need to be accepted, including [self-archiving policies](https://www.nature.com/nature-portfolio/editorial-policies/self-archiving-and-license-to-publish). Those licensing terms will supersede any other terms that the author or any third party may assert apply to any version of the manuscript.

As soon as your article is published, you will receive an automated email with your shareable link